# Hydrodynamic noise in one dimension: projected Kubo formula and how it vanishes in integrable models

Benjamin Doyon

Department of Mathematics, King's College London, Strand, London WC2R 2LS, U.K.

Hydrodynamic noise is the Gaussian process that emerges at larges scales of space and time in many-body systems. It is justified by the central limit theorem, and represents degrees of freedom forgotten when projecting coarse-grained observables onto conserved quantities. It is the basis for fluctuating hydrodynamics, where it appears along with "bare" diffusion terms related to the noise covariance by the Einstein relation. In one dimension of space, nonlinearities are relevant and may modify the corrections to ballistic behaviours by superdiffusive effects. But in systems where no shocks appear, such as linearly degenerate systems and integrable systems, the diffusive scaling of these corrections stays intact. Nevertheless, anomalies remain. We show that in such systems, the noise covariance is given by a modification of the Kubo formula, where effects of ballistic long-range correlations have been projected out. We further show that nonlinearities are tamed by a point-splitting regularisation. With these ingredients, we obtain a well-defined hydrodynamic fluctuation theory in the ballistic scaling of space-time, as a stochastic PDE. It describes the asymptotic expansion in the inverse variation scale of connected correlation functions, self-consistently organised via a cumulant expansion. The resulting anomalous hydrodynamic equation, for average conserved densities, takes into account both long-range correlations and bare diffusion, generalising a recent work by Hübner, Biagetti, De Nardis, and Doyon. Despite these anomalies, two-point functions satisfy an ordinary diffusion equation, with diffusion matrix determined by the Kubo formula. In integrable systems, we show that hydrodynamic noise, hence bare diffusion, must vanish, as was conjectured recently, and argue that under an appropriate gauge of the currents, this is true at all orders. Thus the Ballistic Macroscopic Fluctuation Theory give the all-order hydrodynamic theory for integrable models.

# 1   Introduction

At large scales of space and time, degrees of freedom in many-body systems separate into "fast" and "slow" ones. Fast ones can be assumed to quickly relax throughout space-time, at each time taking values that are determined by the spatial profile of slow ones. This is the projection principle of Mori and Zwanzig [1, 2]. Slow degrees of freedom are associated with conserved quantities [3, 4, 5, 6]: coarse-grained conserved densities vary slowly due to interchanges at the surface of "fluid cells", while generic coarse-grained observables fluctuate a lot, thus converging to their microcanonical values in each fluid cell. At the linearised level, this is called the Boltzmann-Gibbs principle [6], and keeping microcanonical values as nonlinear functions of densities, it is the basis for the Ballistic Macrocopic Fluctution Theory (BMFT) [7, 8], a large-deviation theory.

    This is exact in the strict limit where spatial variations are on infinite length scales $\ell$, as fluid cells can be taken as large as we want. But looking at corrections in the *hydrodynamic expansion* in $1/\ell$, this projection is not perfect. The microcanonical averages on fluid cells receive many corrections. In particular, by the central limit theorem, there must be an emergent Gaussian white noise for each coarse-grained observable. This noise is usually assumed to be delta-correlated in space-time, but correlated amongst different observables, and by the fluctuation-dissipation theorem and Einstein relation, is argued to be associated with diffusion terms [4, 5, 6, 9]. This is the basis for the theory of fluctuating hydrodynamics, a stochastic PDE. It applies both to deterministic systems, where the noise is an emergent phenomenon carrying initial fluctuations, and stochastic systems, where the emergent hydrodynamic noise is in general not simply related to the microscopic noise. Its applicability to many-body quantum systems has also been investigated [10].

In driven-diffusive systems, and under the diffusive scaling of space and time $\vec{x} - \vec{v}t \propto \ell$, $t \propto \ell^2$, the combination of fluctuating hydrodynamics and fluctuations of initial conditions gives rise to the Macroscopic Fluctuation Theory (MFT) [11], a large-deviation theory.

But fluctuating hydrodynamics goes beyond, and holds also when ballistic fluxes – microcanonical averages of currents – are nonlinear functions. For instance, take a system with a single conservation law $\partial_t q + \partial_x j = 0$, where $x, t$ are microscopic coordinates. One writes the coarse-grained current $\bar{j}$ as a function of the coarse-grained conserved density $\bar{q}$

$$\bar{j} = \mathsf{j}(\bar{q}) - \frac{1}{2}\tilde{\mathfrak{D}}(\bar{q})\partial_x\bar{q} + \eta \tag{1}$$

where $\mathsf{j}(\bar{q})$ is the microcanonical average of the current in a state characterised by $\bar{q}$, and $\eta(x, t)$ is an emergent noise, correlated on microscopic space-time scales only, with strength $\tilde{\mathfrak{L}}(\bar{q}) = \tilde{\mathfrak{D}}(\bar{q})\chi(\bar{q})$ where $\chi(\bar{q})$ is the susceptibility. The fluctuating hydrodynamics, valid on large scales of $x, t$, is the stochastic PDE

$$\partial_t\bar{q} + \partial_x\left(\mathsf{j}(\bar{q}) - \frac{1}{2}\tilde{\mathfrak{D}}(\bar{q})\partial_x\bar{q} + \eta\right) = 0. \tag{2}$$

Crucially, the noise strength $\tilde{\mathfrak{L}}$ and bare diffusion $\tilde{\mathfrak{D}}$ are generically different from their macroscopic counterparts [12, 13, 14, 15, 16]. The evaluation of bare transport coefficients is delicate [13, 17, 18, 19, 20], and the understanding of the effect of hydrodynamic noise on large-scale physics is an important question, potentially related to spontaneous stochasticity [21] and the dissipation anomaly in turbulence [22]. For instance, in low-dimensional systems, $\tilde{\mathfrak{L}}$ is not given by the Onsager coefficient – the Green-Kubo formula from the current-current correlator. In fact, the Onsager coefficient often diverges on infinite volumes, indicating the presence of superdiffusion. Notably, if the second derivative $\mathsf{j}''$ is non-vanishing, Eq. (2), under the KPZ scaling of space and time, is known to lie in the KPZ universality class. With more conservation laws, $\partial_t q_i + \partial_x j_i = 0$, $i = 1, 2, \ldots$, the KPZ universality class also emerges [23], as controlled by the "diagonal 3-point couplings", equilibrium 3-point correlation functions $\mathsf{A}_k^{II} = \langle Q_I, Q_I, j_k^-\rangle^{\mathrm{c}}$ simply related to the ballistic flux derivatives [24] (see Appendix A).

Interestingly, one can show that *if the Euler-scale equation is such that shocks are not produced, then the diagonal 3-point couplings vanish, $\mathsf{A}_k^{II} = 0$, thus there is no superdiffusion*, see [25] and Appendix C.1. A class of such no-shock systems is that of *linearly degenerate systems*. Linear degeneracy was first introduce by Lax [26], see also [27, 28, 29] and [30, Sec 3.2]. It means that hydrodynamic modes do not self-interact: the hydrodynamic velocity of any normal mode, does not depend on this normal mode, which precludes the appearance of shocks [31, 32, 30]. Integrable many-body systems have linearly degenerate hydrodynamic equations [28, 29, 33] and are proven not to develop shocks [34, 35], and there are many other examples [36, 37, 38, 39].

But despite the absence of superdiffusion, there are still large-scale anomalies in linearly degenerate systems.

Consider the *the ballistic scaling*

$$x = \ell x, \quad t = \ell t, \quad \ell \to \infty. \tag{3}$$

In no-shock systems, the fluctuating theory at the leading order is simply the deterministic transport of initial fluctuations via the Euler equation: this makes sense as the absence of shocks

makes solutions unique, and entropic effects do not occur. This is the BMFT [7, 8]. But beyond the leading order, noise and diffusion appears: under the scaling (3) we may expect to have non-zero bare contributions $\frac{1}{2\ell}\tilde{\mathfrak{D}}(\overline{q})\partial_x\overline{q} + \frac{1}{\ell}\xi$ in (2), where $\xi$ is delta-correlated in space-time. The question of the exact diffusive scale fluctuating hydrodynamics, bare transport coefficients, and their effects on large-scale correlation functions and the hydrodynamic equation for average densities, has not been addressed. In fact, it was recently conjectured that in integrable systems, there is no bare diffusion neither noise, $\xi = 0$ [36, 37, 25], even though *there are diffusive-scale effects* on the hydrodynamic equation [25], and two-point correlation functions satisfy a diffusive equation [40, 41, 42].

In this paper, we propose the exact diffusive scale fluctuating hydrodynamic theory for no-shock systems in the ballistic scaling (3). Along with a nonlinear fluctuating Boltzmann-Gibbs principle for generic observables, Eq. (21) below, this gives a framework for evaluating the leading and first subleading order in the asymptotic $\ell \to \infty$ expansion of all $n$-point connected correlation functions of local observables, $n = 1, 2, 3, \ldots$. The leading order is given by the BMFT and worked out in [8, 43]. The first subleading order is a $1/\ell$ relative correction. The resulting stochastic PDE, Eq. (26) below, is well defined for this asymptotic expansion. In particular, we evaluate exactly the noise strength and bare diffusion in terms of a modification of the Green-Kubo formula. Our findings are as follows:

(1) We show that there is a correction to the fluid-cell microcanonical average, in inverse power of the cell's size. This correction is usually not discussed in works on fluctuating hydrodynamics, but is crucial. It means that fluxes are subject to a "regularisation", $\mathsf{j}(\overline{q}) \to \mathsf{j}^{\mathrm{reg}}(\overline{q})$, which we show is the point-splitting regularisation first conjectured in [25]. This in turn guarantees that the stochastic PDE is well defined in the $1/\ell$ expansion, despite delta-correlations (in macroscopic coordinates) in initial conditions.

(2) We show that the noise covariance is given by a "projected" version of the Green-Kubo formula. In this version, quadratic charges [24] are projected out. These represent the effects of wave interaction which are at the source of long-range correlations [7], and come from the nonlinearity of the flux as worked out in [25] in the context of integrability. The fact that flux nonlinearities give rise to diffusive contributions was first proposed as "diffusion from convection" in [44] in the context of integrability, and our proof uses a calculation that is somewhat similar.

(3) We show that the bare diffusion matrix is connected to the projected Onsager matrix by the Einstein relation. This relation is natural on the grounds of the fluctuation-dissipation theorem, but it remains an assumption in the context of fluctuating hydrodynamics. We show that it is a consequence of the self-consistency of the PDE as a principle for the $1/\ell$ expansion, along with the conservation laws.

(4) We show that in many-body integrable models, the noise for currents vanishes. This follows from our result, along with earlier calculations of the full Onsager matrix in integrable models [45, 5, 40, 41, 42] (that used different methods). This establishes the conjecture of [36, 37, 25]. This implies that in integrable systems, initial fluctuations do not influence the diffusive corrections to coarse-grained nonlinear currents as functions of conserved densities. We also propose a more heuristic derivation of the vanishing of noise from first principles, using the infinite family of commuting flows that many-body integrable models admit. The argument shows that under an appropriate choice of "gauge" for the currents, their noise vanish at all orders in $1/\ell$. It also effectively gives a new, first-principle derivation of the exact formula for

the full Onsager matrix in integrable models. We emphasise, however, that in generic linearly degenerate systems, the noise does not vanish (contrary to the conjecture made in [25]) – this is a property of integrability.

From our fluctuating hydrodynamic theory, we write down the hydrodynamic equation up to, including, the diffusive order $1/\ell$, see Eqs. (111), (112) below. It is a simple modification of that proposed in [25] for integrable models. Long-range corelations and bare diffusion are explicitly taken into account separately.

Interestingly, the projected Kubo formula for the noise is also worked out, but in the diffusive instead of ballistic scaling, in the simultaneous paper [39] of which we have been made aware at the time of writing. Thus this formula holds no matter the choice of scaling. In particular, in [39] it is evaluated in a specific model, where it is shown to agree with microscopic calculations and to be non-zero.

We emphasise that bare diffusion is different from hydrodynamic diffusion. Linearised hydrodynamics for two-point correlation functions in linearly degenerate systems is not anomalous, and simply diffusive. There, the hydrodynamic diffusion constant is connected by the Einstein relation to the *full Onsager matrix*, not the projected one (see e.g. [9]). We show that this follows from fluctuating hydrodynamics theory; it was also argued for in integrable models in [25] using different ideas, and assumed to be true for some time [40, 41, 42]. Thus, integrable models have normal diffusive corrections to ballistic two-point functions. Beyond the linearised level, however, these diffusive effects desintegrate into long-range correlations [25]; higher-point functions have anomalous diffusive-scale corrections.

We believe our general explanations and precise derivation give a better understanding of the microscopic origin of hydrodynamic noise and bare diffusion, and of the subtle principles behind fluctuating hydrodynamics more generally.

The paper is organised as follows. In Section 2 we specify our general setup, describing the conserved quantities and states. In Section 3, we express our main result for the fluctuating hydrodynamic theory, and provide arguments as to how coarse-graining gives rise to the special effects seen beyond the Euler scale, including the noise. In Section 4, we prove points (1)-(3) described above. In Section 5 we show the absence of noise, point (4) above, in integrable systems. In Section 6, we obtain, from the fluctuating theory, the hydrodynamic equation for one-point averages out of equilibrium, and for two-point correlations in stationary states. Finally, in Section 7 we conclude.

*Note added:* Shortly after the first version of this paper was posted on arXiv, a paper by F. Hübner was also posted [46] which presents an analytical and numerical study of the hard rods model confirming our finding that hydrodymamic noise in integrable systems vanish at the diffusive level.

## 2 Conservation laws and states

Consider a dynamical system in one spatial dimension, which may be deterministic or stochastic, with short-range, translation invariant interactions. Fluctuating hydrodynamics, where a classical noise is assumed to emerge, is naturally a classical theory, and quantum effects may affect this in quantum systems. Many of the results below may still be applicable to quantum models

[10], however we will not discuss this.

In order to have a universal description, one powerful method is to concentrate on *local observables* – such as in the algebraic formulation of statistical mechanics [47, 48]. Roughly, for our purposes, a local observable, denoted $o(\mathrm{x},\mathrm{t})$, is a function on phase space, or a random variable of the stochastic process, etc, which is supported (in a natural sense) on some finite region of space in microscopic units, around position $\mathrm{x} \in \mathbb{R}$ at time $\mathrm{t} \in \mathbb{R}$. Observables can be multiplied with each other (so they form an algebra), and states (see below) are positive linear maps on this algebra.

We will write $\mathrm{x} = \ell x$, $\mathrm{t} = \ell t$, defining the space-time point coordinates $(x, t)$ in macroscopic unit, where $\ell$ is the large macroscopic scale[1].

The system admits a certain number of extensive conserved quantities[2].

$$Q_i = \int \mathrm{dx}\, q_i(\mathrm{x},\mathrm{t}), \quad \partial_{\mathrm{t}} Q_i = 0 \tag{4}$$

where $q_i(\mathrm{x},\mathrm{t})$ is the associated local density. The index $i$ lies in some index set $\mathscr{I}$ which may be finite or not. The index may even be continuous like in integrable models where it represents the spectral space (such as velocities of quasiparticles); we still use the notation $\sum_i$. We assume that there are associated local currents $j_i(\mathrm{x},\mathrm{t})$ giving rise to local conservation laws:

$$\partial_{\mathrm{t}} q_i(\mathrm{x},\mathrm{t}) + \partial_{\mathrm{x}} j_i(\mathrm{x},\mathrm{t}) = 0. \tag{5}$$

Total charges $Q_i$, on infinite volume, do not make sense as observables because they are infinite, but make sense within connected correlation functions, which is how we will use them.

When discussing hydrodynamics at the diffusive order, the concept of PT symmetry plays an important role, as first discussed in [41]. We assume that there is an involution $\mathscr{P}T$ of the algebra of observables that reverses space and time, and that *preserves all conserved densities and their currents*. Further, we will only consider local observables $o$ that are real or hemitian, and *that are preserved by the PT transformation*. That is,

$$\mathscr{P}T(q_i(\mathrm{x},\mathrm{t})) = q_i(-\mathrm{x},-\mathrm{t}), \quad \mathscr{P}T(j_i(\mathrm{x},\mathrm{t})) = j_i(-\mathrm{x},-\mathrm{t}), \quad \mathscr{P}T(o(\mathrm{x},\mathrm{t})) = o(-\mathrm{x},-\mathrm{t}). \tag{6}$$

This simplifies the discussion in subtle ways. Note that in general, there may be observables $o$ with negative eigenvalue under the $\mathscr{P}T$ involution; however, our results do not apply to such observables.

The macrocanonical ensemble is that which maximises entropy under the constraints of the extensive conserved quantities. Formally, it has the Gibbs form

$$\langle \cdots \rangle_{\underline{\beta}} = \lim_{\mathcal{V} \to \mathbb{R}} \frac{\int \mathrm{d}\mu\, \mathrm{e}^{-\sum_i \beta^i Q_i} \cdots}{\int \mathrm{d}\mu\, \mathrm{e}^{-\sum_i \beta^i Q_i}} \tag{7}$$

where $\underline{\beta} = (\beta^1, \beta^2, \dots)$ are the "Lagrange parameters", the charges $Q_i = \int_{\mathcal{V}} \mathrm{d}^d\mathrm{x}\, q_i(\mathbf{x})$ lie on the volume $\mathcal{V}$, and the large-volume limit $\mathcal{V} \to \mathbb{R}$ is assumed to exist for averages of local observables.

---

[1]We consistently use the different fonts x, t and $x, t$ for microscopic and macroscopic coordinates, respectively.

[2]Integrals without domain specification are on the full space, which is $\mathbb{R}$ in continuous sytems or in macroscopic units, and $\mathbb{Z}$ in systems with discrete space and in microscopic units.

$\mu$ is a time-invariant, homogeneous prior measure, for instance in classical Hamiltonian systems with canonical coordinates $(x_a, p_a)$ one chooses $d\mu = \sum_{N=0}^{\infty} (N!)^{-1} \prod_{a=1}^{N} dx_a dp_a$ (direct sum). We assume that $\beta^i$'s are away from phase transitions, so that correlations are short-range in these states, and averages of local observables are smooth in $\underline{\beta}$. Likewise, the microcanonical ensemble is denoted (by a slight abuse of notation)

$$\langle \cdots \rangle_{\underline{\mathsf{q}}} = \lim_{\mathcal{V} \to \mathbb{R}} \frac{\int_{Q_i/|\mathcal{V}| \in [\mathsf{q}_i - \epsilon, \mathsf{q}_i + \epsilon] \forall i} d\mu \cdots}{\int_{Q_i/|\mathcal{V}| \in [\mathsf{q}_i - \epsilon, \mathsf{q}_i + \epsilon] \forall i} d\mu} \tag{8}$$

where $\epsilon \to 0$ as $\mathcal{V} \to \mathbb{R}$ in an appropriate fashion [49, 50]. The covariance matrix,

$$\mathsf{C}_{ij} = -\frac{\partial \langle q_j \rangle_{\underline{\beta}}}{\partial \beta^i} = \langle Q_i, q_j \rangle_{\underline{\beta}}^{\mathrm{c}} = \int dx \, \langle q_i(x), q_j(0) \rangle_{\underline{\beta}}^{\mathrm{c}} = \int dx \left( \langle q_i(x) q_j(0) \rangle_{\underline{\beta}} - \langle q_i(x) \rangle_{\underline{\beta}} \langle q_j(0) \rangle_{\underline{\beta}} \right) \tag{9}$$

is always non-negative, and must be assumed to be positive for conserved quantities to have a hydrodynamic meaning[3]. That is,

$$\mathsf{C}^{\mathrm{T}} = \mathsf{C}, \quad \mathsf{C} > 0, \tag{10}$$

and the map $\underline{\beta} \mapsto \langle \underline{q} \rangle_{\underline{\beta}}$ is invertible. This defines functions $\beta^i(\underline{\mathsf{q}}) = \beta^i(\mathsf{q}_1, \mathsf{q}_2, \ldots)$ for all $i$'s. There is equivalence between microcanonical and macrocanonical ensembles [49, 50, 52]: for every local obsevable $o$, we have $\langle o \rangle_{\underline{\mathsf{q}}} = \langle o \rangle_{\underline{\beta}(\underline{\mathsf{q}})}$. Throughout, we denote the resulting average of $o$, as a function of conserved densities $\underline{\mathsf{q}}$, by using the "sans-serif" font for the observable,

$$\mathsf{o}(\underline{\mathsf{q}}) := \langle o \rangle_{\underline{\mathsf{q}}} = \langle o \rangle_{\underline{\beta}(\underline{\mathsf{q}})}. \tag{11}$$

See Appendix A for hydrodynamic matrices, velocities and normal modes.

For our general discussion of hydrodynamics, we consider states which are not space-time stationary, but with variations on long wavelengths $\ell$. It is useful to think of states of the form

$$\langle \cdots \rangle_{\ell} = \frac{\int d\mu \, e^{-\sum_i \int dx \, \beta^i(x/\ell) q_i(x)} \cdots}{\int d\mu \, e^{-\sum_i \int dx \, \beta^i(x/\ell) q_i(x)}} \tag{12}$$

for a large, macroscopic scale $\ell$. At initial time, this state is a local-equilibrium state,

$$\lim_{\ell \to \infty} \langle o(\ell x, 0) \rangle_{\ell} = \langle o \rangle_{\beta(x)}. \tag{13}$$

It also has short range correlations: for every fixed $x_1, y_1, x_2, y_2 \in \mathbb{R}$, one has

$$\langle o_1(\ell x_1 + y_1, 0), o_2(\ell x_2 + y_2, 0) \rangle_{\ell}^{\mathrm{c}} = \left( \langle o_1(y_1, 0), o_2(y_2, 0) \rangle_{\beta(x_1)}^{\mathrm{c}} + \mathcal{O}(\ell^{-2}) \right) \delta_{x_1, x_2} + \mathcal{O}(\ell^{-\infty}) \tag{14}$$

where $\delta_{x_1, x_2} = 1$ if $x_1 = x_2$ and 0 otherwise. A more precise expression of these initial correlations, as distributions on the macroscopic coordinates, is given in (24) below. At later times it develops long-range correlations [7]. More general states, with long-range correlations or with long-wavelength source insertions at various macroscopic times, can also be considered.

---

[3]This guarantees convexity of the free energy, a fundamental concept of statistical mechanics. See e.g. [51] for a general theorem concerning this in the context of the linearised Euler equation.

# 3 Local relaxation with fluctuations

At the basis of hydrodynamics is the idea of separation of scales: some observables vary and fluctuate slowly in space and time, while other do so much more quickly. By assuming that fast observables relax quickly, there remains a theory for slow observables, which are the local conserved densities admitted by the system. The Boltzmann Gibbs principle for hydrodynamics at the Euler scale says that linear fluctuations of observables project onto those of densities under such a fast relaxation [6]. Its nonlinear version says that every local observable is a fixed, non-fluctuating function of conserved densities, which are themselves fluctuating. This gives access to ballistic-scaling large deviations [7, 8, 43]. Beyond the Euler scale, one must add noise and diffusive effects. This is at the basis fluctuating hydrodynamics.

In this section we write the corresponding nonlinear fluctuating Boltzmann-Gibbs principle, and in order to justify its various parts, provide heuristic arguments for it. We consider the ballistic scaling of space and time (3), and its corrections in $1/\ell$. Most of this discussion is relevant to other scalings as well, such as the diffusive scaling, see [53, 54, 37, 36, 39].

## 3.1 Nonlinear fluctuating Boltzmann-Gibbs principle

A fluid cell is an abstract construct that is useful in order to determine the large-scale properties of many-body system. In one dimension, it is an interval of size $L$ much larger than microscopic lengths $\ell_{\mathrm{micro}}$ but much smaller than variation lengths $\ell$,

$$\ell \gg L \gg \ell_{\mathrm{micro}}. \tag{15}$$

Both $\ell_{\mathrm{micro}}$ (determined by the microscopic model) and $\ell$ (determined by the initial state) are measurable. $L$ is not, but it is a convenient theoretical concept. We may consider $\ell_{\mathrm{micro}}$ to be finite (say 1), by appropriate choice of microscopic units. We write

$$V_L = [-L/2, L/2] \tag{16}$$

and define the fluid-cell mean of an observable – the coarse grained observable – at macroscopic space-time coordinates $x, t$ as

$$\bar{o}(x,t) := \frac{1}{L} \int_{V_L} \mathrm{d}y \, o(\ell x + \mathrm{y}, \ell t). \tag{17}$$

This precise form of the fluid cell is not mandatory; one may for instance consider an even weight function $w(\mathrm{x}) = w(-\mathrm{x})$ such that $|w(\mathrm{x})| < a\mathrm{e}^{-b|\mathrm{x}|}$ for some $a, b > 0$ and $\int_{\mathbb{R}} \mathrm{d}\mathrm{x} \, w(\mathrm{x}) = 1$, and define

$$\bar{o}(x,t) := \frac{1}{L} \int_{\mathbb{R}} \mathrm{d}y \, w(\mathrm{y}/L) o(\ell x + \mathrm{y}, \ell t). \tag{18}$$

In order to simplify the form of the initial correlations in the state (12), it is important that the fluid-cell mean be *balanced*: the fluid-cell averaging is invariant under $\mathrm{y} \to -\mathrm{y}$.

We argue in Sec. 3.2 that a *nonlinear, fluctuating Boltzmann-Gibbs principle* holds. That is, coarse-grained observables can be written solely in terms of fluctuating fields representing coarse-grained conserved densities, which by a slight abuse of notation we denote $q_i(x,t)$, and

emergent Gaussian white noise fields $\xi_o(x,t)$'s representing the "forgotten" fast microscopic degrees of freedom for the observable $o$. Both of these families of fields fluctuate, and we represent averages in the corresponding emergent fluctuating theory as $\langle\!\langle\cdots\rangle\!\rangle_\ell$. The emergent fluctuating theory implicitly depends on $L$ via our choice of fluid cell, but it only gives "universal" results, independent of how $L$ is chosen, in the limit (15).

Before we express this emergent fluctuating theory, we mention that it is expected to describe the large-$\ell$ asymptotic expansion of all connected correlation functions at distinct macroscopic space-time coordinates $(x_i, t_i) \neq (x_j, t_j) \ \forall i \neq j$, as

$$\langle \overline{o_1}(x_1,t_1),\ldots,\overline{o_n}(x_n,t_n)\rangle_\ell^c = \langle\!\langle o_1(x_1,t_1),\ldots,o_n(x_n,t_n)\rangle\!\rangle_\ell^c + \mathcal{O}(\ell^{-1-n}) \tag{19}$$

where $\langle\!\langle o_1(x_1,t_1),\ldots,o_n(x_n,t_n)\rangle\!\rangle_\ell^c$ are distributions, of the form

$$\langle\!\langle o_1(x_1,t_1),\ldots,o_n(x_n,t_n)\rangle\!\rangle_\ell^c = \ell^{1-n}\mathsf{S}_{o_1,\ldots,o_n}(x_1,t_1;\cdots;x_n,t_n) + \ell^{-n}\delta\mathsf{S}_{o_1,\ldots,o_n}(x_1,t_1;\cdots;x_n,t_n) + \ldots \tag{20}$$

The leading large-deviation scaling $\ell^{1-n}$ is explained in [55, 8, 43].

The nonlinear Boltzmann-Gibbs principle says that the random variable $o(x,t)$ in the fluctuating theory $\langle\!\langle\cdots\rangle\!\rangle_\ell$, representing the coarse-grained $\overline{o}(x,t)$, has three parts: microcanonical, diffusive and stochastic (Gaussian white noise)[4]. For all[5] $t > 0$,

$$\boxed{\begin{aligned} &\overline{o}(x,t) \text{ in } \langle\cdots\rangle_\ell \\ &\longrightarrow \quad o(x,t) := \underbrace{\mathsf{o}^{\mathrm{reg}}(\underline{q}(x,t))}_{\text{microcanonical}} - \underbrace{\frac{1}{2\ell}\sum_i \hat{\mathfrak{D}}_o^{\ i}(\underline{q}(x,t))\partial_x q_i(x,t)}_{\text{diffusive}} + \underbrace{\frac{1}{\ell}\xi_o(x,t)}_{\text{noise}} \text{ in } \langle\!\langle\cdots\rangle\!\rangle_\ell \end{aligned}} \tag{21}$$

and in particular, for conserved densities,

$$\overline{q_i}(x,t) \longrightarrow q_i(x,t). \tag{22}$$

For the initial state (12), the noise $\xi_o(x,t)$ is exactly zero at $t = 0$,

$$\xi_o(x,0) = 0, \tag{23}$$

but it is nonzero generically for all $t > 0$, see below. The initial state gives rise to the following one- and two-point correlations for the emergent conserved density fields,

$$\begin{aligned} \langle\!\langle q_i(x,0)\rangle\!\rangle_\ell &= \langle q_i\rangle_{\underline{\beta}(x)} + \mathcal{O}(\ell^{-2}) \\ \langle\!\langle q_i(x,0)q_j(x',0)\rangle\!\rangle_\ell &= \ell^{-1}\mathsf{C}_{ij}(x)\delta(x-x') + \mathcal{O}(\ell^{-3}) \end{aligned} \tag{24}$$

---

[4]In correlation functions with $n \geq 3$, if the Euler hydrodynamics admits shocks, we expect (21) to hold almost everywhere in space-time. The specification "almost everywhere" is because at shocks the local relaxation arguments at its basis do not hold, and this, even at the diffusive order; and for $n \geq 3$, the nonlinear expansion required may be affected by shocks. In no-shock systems, such as with linear degeneracy, it is expected to hold everywhere except at colliding space-time positions. See the discussion in [43].

[5]For $t < 0$, the sign of the diffusive part is positive instead of negative, so in general we must replace $- \longrightarrow -\operatorname{sgn} t$ in front of the diffusive term.

where $C_{ij}(x)$ is the covariance matrix evaluated within the state $\underline{\beta}(x)$. Note that $n$-point functions receive $\mathcal{O}(\ell^{-1})$ relative corrections, PT symmetry and because the fluid-cell mean is balanced, as we show in Appendix B.

The fluctuating theory $\langle\!\langle \cdots \rangle\!\rangle_\ell$ is *completely determined by the initial correlations* $S_{q_{i_1},\dots,q_{i_n}}(x_1,0;\cdots,x_n,0)$ *and* $\delta S_{q_{i_1},\dots,q_{i_n}}(x_1,0;\cdots,x_n,0)$, *the replacement* (21), *and the conservation laws*

$$\partial_t q_i(x,t) + \partial_x j_i(x,t) = 0. \tag{25}$$

Indeed, these then provide the dynamics and hence the correlations of conserved densities at later times: this is through the *nonlinear fluctuating hydrodynamic equation* (here for $t > 0$)

$$\boxed{\partial_t q_i(x,t) + \partial_x\left(\mathsf{j}_i^{\mathrm{reg}}(\underline{q}(x,t)) - \frac{1}{2\ell}\sum_k \hat{\mathfrak{D}}_i^{\,k}(\underline{q}(x,t))\,\partial_x q_k(x,t) + \frac{1}{\ell}\xi_i(x,t)\right) = 0} \tag{26}$$

where we use the simplified notation

$$\hat{\mathfrak{D}}_i^{\,k} := \hat{\mathfrak{D}}_{j_i}^{\,k}, \quad \xi_i := \xi_{j_i}. \tag{27}$$

The replacement (21) along with (26) gives the large-$\ell$ asymptotic expansion (20).

Let us discuss the various elements in (21). The quantities $\hat{\mathfrak{D}}_o^{\,i}(\underline{q})$ are the "bare" diffusion functions for the observable $o$, which account for fluctuation-dissipation effects from the noise, They represent irreversible effects (thus the $\mathrm{sgn}(t)$ factor in general). $o^{\mathrm{reg}}(\underline{q})$ is the finite-size-corrected microcanonical average (11) in the fluid cell. It involves a "regularisation", which is *crucial in order to make sense of singularities that arise because of short-range correlations*, as introduced by the initial state, Eq. (24).

Noise fields $\xi_o(x,t)$ are linear functions of $o$. As discussed in different contexts in [56, 57, 58, 59, 60], physically meaningful noise is to be treated in the Itô convention on time. That is, given $\mathcal{F}(t) = \{q_i(x',t') : x' \in \mathbb{R},\ t' \in [0,t], i \in \mathcal{I}\}$, the noise fields $\xi_o(x,t)$, for all observables $o$ (including currents $j_i$) and for all $x \in \mathbb{R}$, are such that $\xi_o(x,t)\mathrm{d}t = \mathrm{d}w_o(x,t) := w_o(x,t+\mathrm{d}t) - w_0(x,t)$ are centered Gaussian *forward* increment, with

$$\langle\!\langle \mathrm{d}w_o(x,t)\mathrm{d}w_{o'}(x',t)|\mathcal{F}(t)\rangle\!\rangle_\ell = \hat{\mathfrak{L}}_{o,o'}\big(\underline{q}(x,t)\big)\delta(x-x')\mathrm{d}t \tag{28}$$

for some symmetric noise covariance $\hat{\mathfrak{L}}_{o,o'}(\underline{q})$, function of the densities.

Note that by this formula, the noise covariance $\hat{\mathfrak{L}}_{o,o'}(\underline{q})$ is such that for any set of local observables $\mathcal{S}$, the matrix $\hat{\mathfrak{L}}(\underline{q})$ with elements $\hat{\mathfrak{L}}_{o,o'}(\underline{q}) : (o,o') \in \mathcal{S} \times \mathcal{S}$ is non-negative. For $o = q_i$ a conserved density, according to (22) the noise must vanish, $\hat{\mathfrak{L}}_{q_i,q_i}(\underline{q}) = 0$ (therefore $\hat{\mathfrak{L}}_{q_i,o}(\underline{q}) = 0\,\forall o$ by the Cauchy-Schwartz inequality).

This gives the noise fields correlations at equal times. In order to express them at different times, we need to take away the explicit dependence of their covariance on the densities. Indeed, these are fluctuating, and correlate non-trivially with previous histories of noise fields because of the fluctuating hydrodynamic equation (26). For this purpose, fix a set of observables $\mathcal{S}$ not linearly related to each other and with non-vanishing noise. Then $\mathcal{S}$ is a basis for its span and the matrix $\hat{\mathfrak{L}}(\underline{q})$ is strictly positive on this basis. Define

$$\hat{\xi}_o = \sum_{o' \in \mathcal{S}} \left(\sqrt{\hat{\mathfrak{L}}^{-1}}\right)_{oo'}\xi_{o'}. \tag{29}$$

Then the associated increments $\mathrm{d}\hat{w}_o(x,t) := \hat{\xi}_o(x,t)\mathrm{d}t$ satisfy

$$\langle\!\langle \mathrm{d}\hat{w}_o(x,t)\mathrm{d}\hat{w}_{o'}(x',t)|\mathscr{F}(t)\rangle\!\rangle_\ell = \delta_{o,o'}\delta(x-x')\mathrm{d}t, \quad o,o' \in \mathcal{S}. \tag{30}$$

It is natural to assume that these are now independent, delta-correlated noise fields over all times,

$$\langle\!\langle \hat{\xi}_o(x,t)\rangle\!\rangle_\ell = 0, \quad \langle\!\langle \hat{\xi}_o(x,t)\hat{\xi}_{o'}(x',t')\rangle\!\rangle_\ell = \delta_{o,o'}\delta(x-x')\delta(t-t'), \quad o,o' \in \mathcal{S}. \tag{31}$$

In general, correlations $\langle\!\langle \xi_o(x,t)\xi_{o'}(x',t')\rangle\!\rangle_\ell$ at different times will be very non-trivial, and $\xi_o(x,t)$ is non-Gaussian, with non-zero higher cumulants at different times. However, for the purpose of the fluctuating theory as a theory for the leading and first sub-leading order for correlation functions, Eqs. (19), (20), noise averages only occur in evaluating the diffusive $\mathcal{O}(\ell^{-1})$ relative correction. At this order, the cumulant expansion makes the noise covariance non-fluctuating, evaluated on the average densities. Hence, from this viewpoint $\xi_o(x,t)$ may be considered Gaussian, and we have

$$\langle\!\langle \xi_o(x,t)\rangle\!\rangle_\ell = 0, \quad \langle\!\langle \xi_o(x,t)\xi_{o'}(x',t')\rangle\!\rangle_\ell = \hat{\mathfrak{L}}_{o,o'}\left(\langle\!\langle \underline{q}(x,t)\rangle\!\rangle_\ell\right)\delta(x-x')\delta(t-t'). \tag{32}$$

When used in conjunction with the fluctuating hydrodynamic equation (26), this makes the latter not a standard stochastic PDE, but rather a *self-consistent stochastic PDE*, where the noise statistics is fixed by the *averages densities solving this evolution equation in a self-consistent manner*.

The meaning of the various terms in (21), as well as the emergence of the Itô convention, is discussed heuristically in Sec. 3.2. These terms will be evaluated in Sec. 4 by evaluating, with this fluctuating theory, particular equilibrium quantities including the Green-Kubo formula.

**Remark 3.1.** *Eq. (21) is the general form expected in fluctuating hydrodynamics (see e.g. [23]). However, the constitutive functions $\hat{\mathfrak{L}}_{o,o'}(\underline{q})$ and $\hat{\mathfrak{D}}_o^{\ i}(\underline{q})$ representing the corrections to the Euler scale must be evaluated, as well as the regularisation $\mathsf{o}^{\mathrm{reg}}(\underline{q})$, which is typically not discussed. In systems that are purely diffusive, for current observables $o = j_k$ only the diffusive and stochastic part remain, as the microcanonical part is independent of $\underline{q}(x,t)$. This is the basis for the Macroscopic Fluctuation Theory (MFT) [11]. Without regularisation, diffusive and noise term, the result is the Ballistic Macroscopic Fluctuation Theory [8]. These are large-deviation theories, which avoid singularities due to the noise and initial conditions.*

**Remark 3.2.** *Usually one only considers noise terms associated to the local currents, and takes $\mathcal{S} = \{j_i : i \in \mathcal{I}\}$. However, in general, every observable $o$ has its associated, independent noise $\xi_o$, generically correlated to all other observables, including currents. It is simple to see from the evolution equation (26) that noise fields for observables that are not currents decouple from the theory, so may be omitted if one looks only at densities and currents.*

## 3.2 Coarse graining and the origin of hydrodynamic noise

We now provide general heuristic arguments for why (21) holds. Although these general arguments suggest the form of each correction, they do not allow us to compute them. However, Eq. (21), as a principle for the large-$\ell$ asymptotic expansion of connected correlation functions,

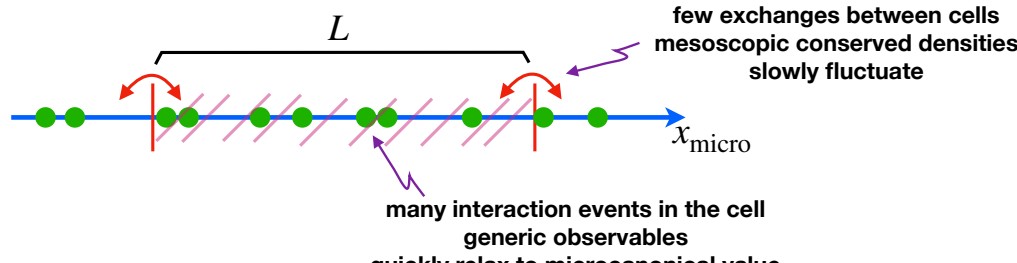

Figure 1: A fluid cell of length $L$ over time $T$. There are any more interactions within the cell than there are boundary effects, leading to a separation of scales between fluctuations of arbitrary local observables and fluctuations of conserved densities.

gives us a framework from which we will evaluate à la Kubo, in the next section, all constitutive elements in terms of equilibrium correlation functions, in linearly degenerate systems.

Recall the fluid-cell mean (17). The main argument is that, because of the conservation laws, fluid-cell means of conserved densities are affected only by boundary effects during the dynamics, while generic observables are affected by every interactions within the volume of the cell. This leads to a separation of scales between conserved densities and generic observables. See Fig. 1.

Let us be more quantitative. By the conservation law, the time derivative of the fluid-cell mean of a conserve density, with respect to time $\mathrm{t} = \ell t$ at position $\mathrm{x} = \ell x$ in microscopic units, is

$$\frac{\partial}{\partial \mathrm{t}} \overline{q_i}(x, t) = L^{-1}\big(j_i(\mathrm{x} - L/2, \mathrm{t}) - j_i(\mathrm{x} + L/2, \mathrm{t})\big). \tag{33}$$

Assuming finite densities and finite typical microscopic velocities, the current observables are of order 1 and vary on a time scale $T$ of order 1 as well. Therefore, $\overline{q_i}(x, t)$ has variations of order $\mathcal{O}(L^{-1})$, and this happen on a time scale of order 1. Physically, these are due to surface effects, such as particles entering or exiting the fluid cell, and interaction between particles through the interface of the cell (which are the points $\mathrm{x} \pm L/2$; such surface effects are small. Then, on $T = \mathcal{O}(1)$, $\overline{q_i}(x, t)$ can be considered to be invariant, and in particular non-fluctuating.

Now consider the fluid-cell mean of an observable that is *not conserved*, $\overline{o}(x, t)$. Because it is composed of $\mathcal{O}(L)$ essentially independent observables, each affected by independent fluctuations, the quantity $\overline{o}(x, t)$ varies on a time scale of order $1/L$, each variation being of order $1/L$. Therefore, $\overline{o}(x, t)$ fluctuates quickly, with $\mathcal{O}(L)$ fluctuations on a time $T$ of order 1. Physcailly, these are due to volume effects: every "interaction event" between particles, such as collision of particles, within the volume of the cell, affect the fluid-cell mean of the observable, and on a time of order 1, there are $\mathcal{O}(L)$ such interaction events. They can be seen as independent fluctuations as they occur at random places within the volume of the cell and change the fluid-cell mean in incoherent ways. Averaging over $T$, which averages over these fluctuations, and considering that conserved densities are invariant and non-fluctuating, the fluid-cell mean of an arbitrary observable should therefore relax to its microcanonical value $\mathsf{o}(\overline{q_1}(x, t), \overline{q_2}(x, t), \ldots)$. See Fig. 2.

In fact, by typicality [61, 62, 63] the average over $T$ can be argued not to be necessary. This

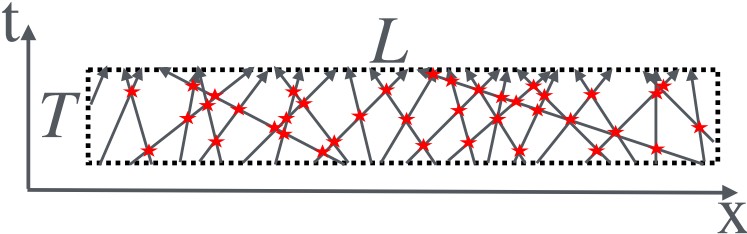

Figure 2: A fluid cell with the typical particle trajectories within it over a time $T$. There are $\mathcal{O}(LT)$ crossings, displayed as red stars, roughly representing independent interactions within this space-time region, while only $\mathcal{O}(T)$ boundary effects.

is because typicality ensures that an average on space implements an average over time in the past, as randomness in the spatial distribution encodes what happened in the past. That is, the $\mathcal{O}(L)$ fluctuations of $\overline{o}(x,t)$ over time $T = \mathcal{O}(1)$ re-organise the system's configuration in $V_L$, under the constraint given by the essentially fixed conserved densities. This re-organisation is enough to implement the small variation of the conserved densities and update $\overline{o}(x, t + T/\ell)$ so that it always take its microcanonical value determined by the $\overline{q_i}(x,t)'s$.

This suggests that

$$\overline{o}(x,t) \stackrel{\ell \gg L \gg \ell_{\text{micro}}}{\rightarrow} \mathsf{o}(\underline{\overline{q}}(x,t)) = \mathsf{o}(\overline{q_1}(x,t), \overline{q_2}(x,t), \ldots). \tag{34}$$

This is the hypothesis of *local relaxation of fluctuations* [7, 8]. It is a nonlinear Boltzmann-Gibbs principle, where fluctuating local observables keep their full nonlinear dependence on the conserved densities.

The above can only be true for $\ell$ large enough. There are two concepts at play. The larger the mesoscopic length $L$ (here we follow the numbering below): (1) the closer is the average itself to the infinite-volume microcanonical average, which equals the macrocanonical average under the right identification of generalised temperatures (Sec. 2); and (3) the more fluctuations are taken into account, hence the less random corrections there are. Yet the cell size cannot be larger than the macroscopic variation scale $\ell$, because the relaxation argument made above requires spatial homogeneity.

Similarly, the larger the time $T$, the more accurate is microcanonical relaxation as well, because more re-organisation has occurred, and also (2) the less there is a time delay between the effective time at which relaxation has occured, and the time at which the values of the conserved densities are taken to determine the state towards which relaxation occurs. But $T$ must not bee too large so that conserved densities can be considered fixed and non-fluctuating.

Our conjecture is that the optimal choice is obtained for

$$LT/\ell_{\text{micro}} = \mathcal{O}(\ell). \tag{35}$$

Note that neither $L$ nor $T$ are univeral quantity – they are artefact of our fluid-cell construction. But $\ell$ is physical – it is the physical variation length scale. Below we often simply set $\ell_{\text{micro}} = 1$, which we can do without loss of generality – this sets the scale of our microscopic coordinates.

For instance, $L = \ell^{1-\epsilon}$ and $T = \ell^\epsilon$ for $\epsilon > 0$ small. The time $T$ is not finite but very small on the scale $\ell$, so the asymptotic expansion as $\ell \to \infty$ introduces small mistakes of order $\epsilon$. The limit $\epsilon \to 0$ is then taken after the asymptotic expansion as $\ell \to \infty$. Another example is $L = \epsilon\ell$ and $T = 1/\epsilon$. Here it is the fluid cell that is too large, introducing mistakes of order $\epsilon$, so that, again, the limit $\epsilon \to 0$ is taken after the asymptotic expansion in $\ell$. This is what is meant by the limit $\ell \gg L \gg \ell_{\mathrm{micro}}$.

What are the corrections at order $1/\ell$ or $1/L$ beyond the $\ell \to \infty$ limit? The argument above suggests that there are three potential corrections.

*1. Corrections to the microcanonical average, $L < \infty$.* Because the cell size is not infinite, the exact average at fixed conserved densities is not the infinite-volume microcanonical average. Hence on the right-hand side of (34) there is a correction

$$+ L^{-1}\delta\mathsf{o}(\overline{q}(x,t)). \tag{36}$$

We gather this into

$$\mathsf{o}^{\mathrm{reg}}(\overline{q}(x,t)) := \mathsf{o}(\overline{q}(x,t)) + L^{-1}\delta\mathsf{o}(\overline{q}(x,t)). \tag{37}$$

This is the finite-$L$ microcanonical average up to, including, $L^{-1}$ corrections. We will see in Sec. 4.1 that these corrections are indeed of order $L^{-1}$. The result appears to be non-universal; as mentioned, contrary to $\ell$, the fluid cell size $L$ is not physical but simply an artefact of our construction. However, the emergent fluctuation theory $\langle\!\langle \cdots \rangle\!\rangle_\ell$ implicitly depends on $L$ because it is a theory for fluctuations of fluid-cell means, and we will show in Sec. 4.1 how, in linearly degenerate systems, these $L$-dependences effectively cancel out and the result is, in fact, universal.

*2. Time delay, $T > 0$: diffusion term.* According to the argument above, there is a delay between the slow variations of the densities and relaxation, so that it is $\overline{o}(x, t + T/\ell)$ that takes the microcanonical value $\mathsf{o}^{\mathrm{reg}}(\overline{q}(x,t))$. The $\ell^{-1}$ expansion gives an additional term. This is expected to be proportional to the spatial variations $\partial_{\mathrm{x}}\overline{q}_i(x,t) = \ell^{-1}\partial_x\overline{q}_i(x,t)$, because $L$ must be taken smaller if the variations are larger, thus $T$ must be taken larger to keep $LT$ the same. Hence on the right-hand side of (34) there is another correction, for $t > 0$:

$$-\frac{1}{2}\ell^{-1}\sum_i \hat{\mathfrak{D}}_o^{\ i}\big(\overline{q}(x,t)\big)\, \partial_x\overline{q}_i(x,t) \tag{38}$$

for some set of diffusion functions $\hat{\mathfrak{D}}_o^{\ i}(\overline{q}(x,t))$ associated to the observable $o$, where the factor $-\frac{1}{2}$ is conventional. For evolution backwards in time, $t < 0$, we use PT symmetry, giving the factor $\mathrm{sgn}\, t$ as both $o$ and $q_i$'s are assumed to be PT invariant. Note how the direction of time is fundamental in this part of the argument: there is a *delay* in reaching the microcanonical average.

*3. Emergent noise, $LT < \infty$.* Microcanonical relaxation, at finite $L$, receives stochastic corrections according to the central limit theorem. Intuitively, this is the result of partially remembering the microscopic degrees of freedom that have been projected out when projecting onto the conserved densities in the nonlinear Boltzmann-Gibbs principle. Hence on the right-hand side of (34) there is a noise correction,

$$\ell^{-1}\xi_o(x,t) \tag{39}$$

with the set of $\xi_o(x,t)$'s, for all local observables $o$'s, being centered Gaussian white noises, delta-correlation in space-time. We need to explain three aspects: (a) why the scaling is $\ell^{-1}$; (b) why the emergent noise is delta-correlated in space-time; and (c) why it is the Itô convention that must be used.

(a) The scaling in $\ell^{-1}$ is obtained as follows. The time-averaged fluid cell over $L \times T$ has $LT = \mathcal{O}(\ell)$ independent fluctuations, therefore a variance due to the noise of order $\mathcal{O}(\ell^{-1})$. Let us write a stochastic contribution $\xi_o^{\mathrm{micro}}(\mathrm{x},\mathrm{t})$ in microscopic coordinates, representing the roughness in both space and time. This can be chosen centered, however it is not Gaussian because it encodes small-scale effects, and it is not a white noise because there are correlations on lengths $\mathcal{O}(\ell_{\mathrm{micro}}) = \mathcal{O}(1)$. We now argue that its two-point correlations can be taken as $\langle\!\langle \xi_o^{\mathrm{micro}}(\mathrm{x},\mathrm{t})\xi_{o'}^{\mathrm{micro}}(\mathrm{x}',\mathrm{t}')\rangle\!\rangle_\ell = \hat{\mathfrak{L}}_{o,o'}\delta_{\ell_{\mathrm{micro}}}(\mathrm{x}-\mathrm{x}',\mathrm{t}-\mathrm{t}')$ for some $\hat{\mathfrak{L}}_{o,o'}$, with mollified space-time delta-function on scale the linear $\ell_{\mathrm{micro}}$. Indeed, with this, the variance of its mean over a fluid cell in space-time is then $\mathcal{O}(\ell^{-1})$, as it should:

$$\mathrm{var}\Big(\frac{1}{LT}\int_{-L/2}^{L/2}\mathrm{dx}\int_0^T\mathrm{dt}\,\xi_o^{\mathrm{micro}}(\mathrm{x},\mathrm{t})\Big) \sim \frac{1}{LT}\hat{\mathfrak{L}}_{o,o'} = \mathcal{O}(\ell^{-1}) \qquad (\ell\to\infty). \tag{40}$$

A similar calculation holds for covariances. Transforming to macroscopic coordinates $\delta_{\ell_{\mathrm{micro}}}(\ell x - \ell x', \ell t - \ell t') \to \ell^{-2}\delta(x-x')\delta(t-t')$, we write $\xi_o^{\mathrm{micro}}(\ell x, \ell t) = \ell^{-1}\xi_o(x,t)$ and we obtain (39) with (32), and on that scale the noise becomes Gaussian as higher-point correlations are expected to vanish faster with $\ell^{-1}$.

(b) The delta-correlation in space-time is argued for as follows. It is known that correlations of physical observables in space (out of equilibrium) and in time (in and out of equilibrium) can be strong, and that this is due to hydrodynamic mode propagation. Since hydrodynamic modes are waves of conserved densities, such strong correlations are destroyed when fixing the space-time configuration of conserved densities on every fluid cell. There remains only short-range correlations on scale $\ell_{\mathrm{micro}}$, thus giving $\langle\!\langle \xi_o^{\mathrm{micro}}(\mathrm{x},\mathrm{t})\xi_{o'}^{\mathrm{micro}}(\mathrm{x}',\mathrm{t}')\rangle\!\rangle_\ell = \hat{\mathfrak{L}}_{o,o'}\delta_{\ell_{\mathrm{micro}}}(\mathrm{x}-\mathrm{x}',\mathrm{t}-\mathrm{t}')$.

(c) Finally, the Itô convention is again an effect of time delay. Fluctuations occur on a time following that at which the essentially fixed fluid-cell means of conserved densities are taken. Thus (28) holds, where Gaussian forward increments have a variance that is determined by the conserved densities at the initial time of the increment. Likewise (32) holds thanks to this.

**Remark 3.3.** *In stochastic systems, the emergent noise can often be obtained in a mathematically accurate fashion. But such an understanding is still lacking in general in Hamiltonian systems.*

## 4   Constitutive relations

We now evaluate the beyond-Euler components of the nonlinear fluctuating Boltzmann-Gibbs principle (21). For this purpose, we evaluate various equilibrium correlation functions. It turns out that in order to evaluate correlation functions directly using (21), it is sufficient to know the asymptotic form (20), and to use the cumulant expansion principles from Malyshev's formula (see [43]). In particular, the Euler-scale projection formulas from [8, 43] for 2- and 3-point functions are sufficient for quantities in (21) with a factor of $\ell^{-1}$.

The results below hold in $d = 1$ "no-shock" systems which are strictly hyperbolic. Strict hyperbolicity means that all hydrodynamic velocities are distinct. Further, a no-shock system

is a many-body system whose Euler-scale equation does not develop shock in non-equilibrium situations. In such systems the solutions to Euler equations are unique at all times, and nonlinear Euler-scale projections are valid everywhere in space-time [43]. The most important examples (perhaps the only ones!) of such systems are *linearly degenerate systems* [26, 27, 28, 29, 30], as indeed these do not develop shocks [31, 32, 30]. See Appendix C for the definition of such systems. It is also known that no shocks are produced in integrable systems [34, 35], which are, in a formal sense, also linearly degenerate [33].

We show in Appendix C that, in no-shock systems, the following results hold:

1. Correlation functions of conserved densities keep their local-equilibrium form at later time even in long-wavelength non-equilibrium states such as those emanating from (12), at the Euler scale:

$$\langle\!\langle q_{i_1}(x_1, t), q_{i_2}(x_2, t)\rangle\!\rangle_\ell^c = \ell^{-1}\Big(\mathsf{C}_{i_1, i_2}(x_1, t)\delta(x_1 - x_2) + \text{regular}\Big) + \mathcal{O}(\ell^{-2}). \tag{41}$$

Here the equilibrium covariance matrix is given in (9), and it is to be evaluated at the state characterised by $\mathsf{q}(x_1, t) = \langle\!\langle q(x_1, t)\rangle\!\rangle_\ell$. This is consistent with the initial condition (24). A partial proof for Eq. (41) was given using the BMFT in [8], and a related calculation in [25, Suppl Mat]. We provide a complete proof in Appendix C, which requires strict hyperbolicity.

2. For every local observable, the three-point couplings, Eq. (137), in space-time stationary, maximal entropy states vanish for normal modes of equal velocities:

$$\langle Q_I, Q_{I'}, o_1^-\rangle_{\underline{\beta}}^c = 0 \quad \text{if } \mathsf{v}_I = \mathsf{v}_{I'}. \tag{42}$$

Thus, as we assume strict hyperbolicity, vanishing occurs whenever $I \neq I'$ (but the more general result shown here holds also without strict hyperbolicity). This implies that there is no superdiffusion [23, 24], and was shown from linear degeneracy in [25], Supplementary Material Sec. "Degenerate three-point coupling from linear degeneracy". For readability, we reproduce this proof in Appendix C.

Using these properties of no-shock systems, we will show:

I. **Point-splitting regularisation.** The microcanonical regularisation $\mathsf{o}^{\text{reg}}(\underline{q}(x, t))$ is given by the *point-splitting prescription* introduced in [25],

$$\Big[q_{i_1}(x, t)\cdots q_{i_k}(x, t)\Big]^{\text{reg}} = \frac{1}{k!}\sum_{\sigma \in S(k)} q_{i_1}(x + \sigma_i 0^+, t)\cdots q_{i_k}(x + \sigma_k 0^+, t) \tag{43}$$

where $S(k)$ is the set of permutations of $(1, 2, \ldots, k)$, and $0^+$ is a fixed, positive infinitesimal number. Note how this is a sum over various orderings in space, with infinitesimal displacements.

II. **Projected Kubo formula.** The noise covariance coefficient is given by the *projected Kubo formula*: with $\langle\cdots\rangle = \langle\cdots\rangle_{\underline{q}}$ the space-time stationary maximal-entropy state with averages $\langle q_i\rangle = \mathsf{q}_i$,

$$\hat{\mathfrak{L}}_{o_1, o_2}(\underline{\mathsf{q}}) = \mathfrak{L}_{o_1, o_2}(\underline{\mathsf{q}}) - \frac{1}{2}\sum_{I, I': \mathsf{v}_I \neq \mathsf{v}'_I}\frac{\langle o_1^-, Q_I, Q_{I'}\rangle^c \langle o_2^-, Q_I, Q_{I'}\rangle^c}{|\mathsf{v}_I - \mathsf{v}_{I'}|}. \tag{44}$$

Here $Q_I$ are the total conserved quantities in the normal mode basis (see (142)), $\mathsf{v}_I$ are the associated hydrodynamic velocities, and $\mathfrak{L}_{o_1,o_2}$ is given by the conventional Green-Kubo formula

$$\mathfrak{L}_{o_1,o_2}(\underline{\mathsf{q}}) := \int_{-\infty}^{\infty} dt \left( \int dx \, \langle o_1(\mathrm{x},\mathrm{t}), o_2(0,0) \rangle^{\mathrm{c}} - \mathsf{D}_{o_1,o_2} \right) \tag{45}$$

with "Drude weight"

$$\mathsf{D}_{o_1,o_2} := \lim_{\mathrm{t}\to\pm\infty} \int dx \, \langle o_1(\mathrm{x},\mathrm{t}), o_2(0,0) \rangle^{\mathrm{c}} = \sum_{i_1,i_2} \frac{\partial \mathsf{o}_1}{\partial \mathsf{q}_{i_1}} \frac{\partial \mathsf{o}_2}{\partial \mathsf{q}_{i_2}} \mathsf{C}_{i_1,i_2} \tag{46}$$

where the last equality follows from a general theorem [51]. Note that the quantity $\mathfrak{L}_{ik} := \mathfrak{L}_{j_i,j_k}$ is the usual Onsager matrix.

III. **(Extended) Einstein relation.** The diffusion coefficients $\hat{\mathfrak{D}}_o^{\ k}(\underline{\mathsf{q}})$ are related to the noise covariance by an *Einstein relation*, here extended to diffusion for arbitrary observables $o$ instead of only currents,

$$\hat{\mathfrak{L}}_{o,j_i}(\underline{\mathsf{q}}) = \sum_k \hat{\mathfrak{D}}_o^{\ k}(\underline{\mathsf{q}}) \mathsf{C}_{ki}, \quad \hat{\mathfrak{D}}_o^{\ k}(\underline{\mathsf{q}}) = \sum_i \hat{\mathfrak{L}}_{o,j_i}(\underline{\mathsf{q}}) \mathsf{C}^{ik}. \tag{47}$$

It is also instructive to evaluate the Drude weight (46) within our formalism, in order to confirm that it is consistent. This is done in Appendix D.

Our derivation of Point I below gives a full justification of the point-splitting regularisation that was proposed in [25].

Our main result is Point II, the projected Kubo formula. Our proof uses a calculation that follows somewhat that done in [44]: the "diffusion from convection" term is subtracted. Crucially, the subtraction corresponds to projecting out the quadratic charges forming the "scattering subspace" of the "diffusive space" constructed in [24], $\mathcal{H}_{\mathrm{scat}} \subset \mathcal{H}_{\mathrm{dif}}$. The diffusive space is a Hilbert space constructed with respect to the sesquilinear form given by the Green-Kubo formula

$$(o_1, o_2)_{\mathrm{dif}} := \mathfrak{L}_{o_1,o_2}(\underline{\mathsf{q}}) = \int_{-\infty}^{\infty} dt \left( \int dx \, \langle o_1(\mathrm{x},\mathrm{t}), o_2(0,0) \rangle^{\mathrm{c}} - \mathsf{D}_{o_1,o_2} \right) \tag{48}$$

(recall that our observables $o_1, o_2$ are assumed to be real or Hermitian, however this is easily extended by sesquilinearity). That is, the space $\mathcal{H}_{\mathrm{dif}}$ is obtained by a Gelfand-Naimark-Segal construction: it is easy to show that $(o_1, o_2)_{\mathrm{dif}}$ is positive semi-definite, so its null space is moded out and the resulting space of equivalence classes is completed with respect to the norm $||\cdot||_{\mathrm{dif}} = \sqrt{(\cdot,\cdot)_{\mathrm{dif}}}$. The scattering subspace is that spanned by particular quadratic elements parametrised by two normal mode indices $I, J$:

$$\mathcal{H}_{\mathrm{scat}} := \mathrm{span}\Big\{ w_{IJ} = \lim_{X\to\infty} \frac{|\mathsf{v}_I - \mathsf{v}_J|}{2X} \int_{-X}^{X} dx \, q_I(\mathrm{x}) q_J \Big\} \subset \mathcal{H}_{\mathrm{dif}}. \tag{49}$$

The fact that these lie within $\mathcal{H}_{\mathrm{dif}}$ is nontrivial and argued for in [24]. With the projection $\mathbb{P}_{\mathrm{scat}} : \mathcal{H}_{\mathrm{scat}} \to \mathcal{H}_{\mathrm{dif}}$, it turns out that the second term in (44) is exactly [24]

$$(\mathbb{P}_{\mathrm{scat}} o_1, \mathbb{P}_{\mathrm{scat}} o_2) = \frac{1}{2} \sum_{I,I':\mathsf{v}_I \neq \mathsf{v}'_I} \frac{\langle o_1^-, Q_I, Q_{I'} \rangle^{\mathrm{c}} \langle o_2^-, Q_I, Q_{I'} \rangle^{\mathrm{c}}}{|\mathsf{v}_I - \mathsf{v}_{I'}|}. \tag{50}$$

That is,

$$\hat{\mathfrak{L}}_{o_1,o_2} = \big((1 - \mathbb{P}_{\text{scat}})o_1, (1 - \mathbb{P}_{\text{scat}})o_2\big)_{\text{dif}}. \tag{51}$$

In [24], the subspace $\mathcal{H}_{\text{scat}}$ is physically interpreted as describing the aggregated effect of nonlinear two-wave scattering events of hydrodynamic modes emanating from initial fluctuations and transported by the Euler equation. Nonlinear scattering of deterministically transported fluctuations is the source of long-range correlations, as explained in [7]. As these have $\mathcal{O}(\ell^{-1})$ strength (see (20)), they contribute to $\mathfrak{L}_{o_1,o_2}$ at the same order as the correlation between hydrodynamic noise $\xi_{o_1}$ and $\xi_{o_2}$. This is why this subspace must be subtracted to obtain the noise covariance. Thus, the present result finally puts together these various concepts in a coherent theory.

Point III is expected – for current observables $o = j_k$, the diffusion functions $\hat{\mathfrak{D}}_k{}^i(\underline{q}(x,t))$ (using the notation (27)) are usually assumed to be related to the noise covariance by the Einstein relation. However, as far as we are aware, this has not been derived before in the context of (21).

We believe the results of Points II and III still hold beyond linearly degenerate systems, with an appropriate projection including certain unbounded elements of the scattering Hilbert space that makes the resulting noise covariance finite. However we do not know of a derivation yet.

We note that in [39], the same projected Kubo formula (44) is obtained for linearly degenerate systems by using a fluctuation principle of the type (21) but under the diffusive scaling, instead of the large-deviation scaling. Importantly, the result is verified by explicit calculations in a stochastic model, thus confirming the projected Kubo formula. The calculation methods are quite different, because of the different scaling.

## 4.1 Correction to the microcanonical average and point splitting

Let us consider the microcanonical correction $\delta\mathsf{o}(\underline{q}(x,t))$, Eq. (36), for one-point averages generically in non-equilibrium states. It can be evaluated by studying the average of $o = o(0,0)$ in a stationary, maximal entropy state $\langle \cdots \rangle_{\underline{\beta}}$, and taking into account that it has short-range correlations. Recall that by the equivalence of ensembles,

$$\mathsf{o}(\underline{\mathsf{q}}) = \langle o \rangle_{\underline{\beta}(\underline{\mathsf{q}})} \tag{52}$$

where the left- (right-)hand side is the microcanonical (macrocanonical) ensemble. Here the Lagrange parameters $\beta^i = \beta^i(\underline{\mathsf{q}})$ are functions of the average conserved densities defined by $\mathsf{q}_i = \langle q_i \rangle_{\underline{\beta}(\underline{\mathsf{q}})} = \langle\!\langle q_i \rangle\!\rangle_{\underline{\beta}(\underline{\mathsf{q}})}$. Let us denote for lightness of notation $\langle \cdots \rangle = \langle \cdots \rangle_{\underline{\beta}(\underline{\mathsf{q}})}$ and likewise $\langle\!\langle \cdots \rangle\!\rangle = \overline{\langle\!\langle \cdots \rangle\!\rangle}_{\underline{\beta}(\underline{\mathsf{q}})}$. Then

$$\mathsf{o}(\langle\!\langle \underline{q} \rangle\!\rangle) = \mathsf{o}(\underline{\mathsf{q}}) = \langle o \rangle = \langle \overline{o} \rangle = \langle\!\langle \mathsf{o}(\underline{q}) \rangle\!\rangle + \frac{1}{L}\langle\!\langle \delta\mathsf{o}(\underline{q}) \rangle\!\rangle \tag{53}$$

where we used homogeneity to write the average as an average of a fluid-cell mean, and then used (21) and the vanishing of noise and diffusive parts in a maximal entropy state. We find that the correction term should simply "allow" for the averaging to pass inside the function $\mathsf{o}(\underline{q})$. Recall that $\langle\!\langle \cdots \rangle\!\rangle$ implicitly depends on $L$ because the fluctuating fields $q_i(x,t)$ represents fluctuating fluid-cell means. Let us evaluate equal-point connected correlation functions of these fluctuating

fields using their fluid-cell mean definition. Assuming short-range correlations,

$$
\begin{aligned}
\langle\!\langle q_{i_1}(0,0), \cdots, q_{i_k}(0,0) \rangle\!\rangle^{\mathrm{c}} &= \frac{1}{L^k} \int_{(V_L)^{\times k}} \prod_{i=1}^{k} \mathrm{dx}_i \, \langle q_{i_1}(\mathrm{x}_1,0) \cdots q_{i_k}(\mathrm{x}_k,0) \rangle^{\mathrm{c}} \\
&= \frac{1}{L^{k-1}} \langle Q_{i_1}, \ldots, Q_{i_{k-1}}, q_{i_k} \rangle^{\mathrm{c}}
\end{aligned}
\tag{54}
$$

where corrections (not written) are exponentially small as $L \to \infty$. Therefore we may use a cumulant expansion to get

$$
\begin{aligned}
\langle\!\langle \mathsf{o}(\underline{q}) \rangle\!\rangle &= \mathsf{o}(\underline{\mathsf{q}}) + \frac{1}{2} \sum_{ij} \frac{\partial^2 \mathsf{o}(\underline{\mathsf{q}})}{\partial \mathsf{q}_i \partial \mathsf{q}_j} \langle\!\langle q_i(0,0) q_j(0,0) \rangle\!\rangle^{\mathrm{c}}_{\underline{\beta}} + \mathcal{O}(L^{-2}) \\
&= \mathsf{o}(\underline{\mathsf{q}}) + \frac{1}{2L} \sum_{ij} \frac{\partial^2 \mathsf{o}(\underline{\mathsf{q}})}{\partial \mathsf{q}_i \partial \mathsf{q}_j} \mathsf{C}_{ij} + \mathcal{O}(L^{-2})
\end{aligned}
\tag{55}
$$

where we used (9). Again by the cumulant expansion we have

$$
\frac{1}{L} \langle\!\langle \delta \mathsf{o}(\underline{q}) \rangle\!\rangle = \frac{1}{L} \delta \mathsf{o}(\underline{\mathsf{q}}) + \mathcal{O}(L^{-2})
\tag{56}
$$

and hence from (52) we identify

$$
\delta \mathsf{o}(\underline{\mathsf{q}}) = -\frac{1}{2} \sum_{ij} \frac{\partial^2 \mathsf{o}(\underline{\mathsf{q}})}{\partial \mathsf{q}_i \partial \mathsf{q}_j} \mathsf{C}_{ij}.
\tag{57}
$$

That is, the correction term simply gets rids of the effects of short-range correlations, the singularities arising at equal positions.

In general evaluating correlation functions using (37) with (57) is rather complicated: we need to keep track of all $1/L$ singularities of $\mathsf{o}(\underline{q}(x,t))$, and all corrections $L^{-1}\delta\mathsf{o}(\underline{q})$. These should cancel each other to give something universal (independent of $L$). However, there is a trick that simplifies this procedure and shows universality.

Consider a more general non-equilibrium state $\langle \cdots \rangle_\ell$ such as (12), and recall the property (41). Then a similar calculation as (54) and (55) shows that, in every such state, $L^{-1}\delta\mathsf{o}(\underline{q})$ cancels the $1/L$ singularities of $\mathsf{o}(\underline{q}(x,t))$. In this calculation, Eq. (41) is used only to evaluate the leading $1/L$ correction, hence we can neglect the $1/\ell$ correction in the delta-function coefficient in (41). But we can implement this cancellation simply as

$$
\langle\!\langle \mathsf{o}(\underline{q}) \rangle\!\rangle_\ell + \frac{1}{L} \langle\!\langle \delta \mathsf{o}(\underline{q}) \rangle\!\rangle_\ell = \langle\!\langle \mathsf{o}^{\mathrm{reg}}(\underline{q}) \rangle\!\rangle_\ell
\tag{58}
$$

by defining the regularisation as acting, on monomial of conserved densities at equal space-time points, by avoiding the "fat diagonal" region where coordinates are at microscopic distances from each other:

$$
\left[ \overline{q}_{i_1}(x,t) \cdots \overline{q}_{i_k}(x,t) \right]^{\mathrm{reg}} = \frac{1}{L^k} \int_{(V_L)^{\times k}_{\neq}} \prod_{j=1}^{k} \left( \mathrm{dx}_j \, q_{i_j}(\ell x + \mathrm{x}_i, \ell t) \right)
\tag{59}
$$

with

$$(V_L)^{\times k}_{\neq} = \{(x_1, \ldots, x_k) \in (V_L)^{\times k} : |x_a - x_b| \gg \ell_{\text{micro}} \, \forall a \neq b\}. \tag{60}$$

Indeed, it is the integration on this fat diagonal that gives the $1/L$ singularity of (54). But, out of equilibrium, inside any sector of a given ordering of spatial coordinates, $x_{\sigma_1} > \cdots > x_{\sigma_k}$ for some $\sigma \in P(k)$, correlations are continuous at the Euler scale, according to BMFT (see e.g. [34, 25]). As these correlations give $\ell^{-1}$ subleading corrections to the representation of the observable $o$ within the fluctuating theory, they are sufficient to understand the diffusive scale. Therefore, we may replace the regularised fluid-cell mean (59), where the fat diagonal is omitted, simply by the point-splitting regularisation (43).

Finally, as we can reproduce correlation functions by functional differentiations with respect to external fields in such states (see [55, 43]), this regularisation works for correlation functions as well. This is a universal procedure: the explicit $L$ dependence has disappeared, and all equal-point singularities are avoided.

Therefore, we have derived the point-splitting regularisation (43) conjectured in [25] from the finite-size correction to the microcanonical averaging in fluid cells.

## 4.2 Noise correlations via projected Kubo formula

We now evaluate the correlation matrix $\hat{\mathfrak{L}}_{o,o'}$ in (32). We will show that it is not the usual Kubo formula, but instead a projected Kubo formula, where the quadratic charge subspace [24] has been projected out.

For this purpose, let us consider again a stationary, maximal entropy state $\langle \cdots \rangle$, and evaluate, using (21), the Kubo formula

$$\begin{aligned}
\mathfrak{L}_{o_1,o_2} &= \int_{-\infty}^{\infty} \mathrm{dt} \left( \int \mathrm{dx} \, \langle o_1(\mathrm{x},\mathrm{t}), o_2(0,0) \rangle^{\mathrm{c}} - \mathsf{D}_{o_1,o_2} \right) \\
&= \ell \int_{-\infty}^{\infty} \mathrm{dt} \left( \ell \int \mathrm{dx} \, \langle\!\langle o_1(x,t), o_2(0,0) \rangle\!\rangle^{\mathrm{c}} - \mathsf{D}_{o_1,o_2} \right).
\end{aligned} \tag{61}$$

In the second line we have written it in terms of the emergent fluctuating theory, with scaled space-time coordinates, where we will use (21). We note that the Drude weight (46) can be written as

$$\mathsf{D}_{o_1,o_2} = \ell \sum_{i_1,i_2} \frac{\partial \mathsf{o}_1}{\partial \mathsf{q}_{i_1}} \frac{\partial \mathsf{o}_2}{\partial \mathsf{q}_{i_2}} \int \mathrm{dx} \, \langle\!\langle q_{i_1}(x,t), q_{i_2}(0,0) \rangle\!\rangle^{\mathrm{c}} \tag{62}$$

for every $t \in \mathbb{R}$, therefore we may write

$$\mathfrak{L}_{o_1,o_2} = \ell^2 \int_{-\infty}^{\infty} \mathrm{dt} \int \mathrm{dx} \, \langle\!\langle o_1^-(x,t), o_2^-(0,0) \rangle\!\rangle^{\mathrm{c}} \tag{63}$$

in terms the observable $o$ from which we project out its overlap with the conserved densities:

$$o^- = o - \sum_{i,j} \langle o, Q_i \rangle^{\mathrm{c}} \mathsf{C}^{ij} q_j = o - \sum_i \frac{\partial \mathsf{o}}{\partial \mathsf{q}_i} q_i. \tag{64}$$

Using (21), we analyse the terms that contribute to $\langle\!\langle o_1(x_1,t_1), o_2(x_2,t_2) \rangle\!\rangle^{\mathrm{c}}$ at orders $\ell^{-1}$ and $\ell^{-2}$; and in particular, those that contribute to (61) (equivalently (63)), hence that are

nonzero under space-time integration. For each of the observables $o_1$ and $o_2$, we consider the microcanonical, diffusive and noise terms, and evaluate their correlations.

First, the microcanonical-microcanonical correlation must be evaluated to leading and first subleading order in $\ell^{-1}$, by the cumulant expansion (or Malyshev's formula), see the principles explained in [43]. The result is

$$
\langle\!\langle \mathsf{o}_1^{\mathrm{reg}}(\underline{q}(x_1,t_1)), \mathsf{o}_2^{\mathrm{reg}}(\underline{q}(x_2,t_2)) \rangle\!\rangle^{\mathrm{c}} \tag{65}
$$
$$
\sim \sum_{i_1,i_2} \frac{\partial \mathsf{o}_1}{\partial \mathsf{q}_{i_1}} \frac{\partial \mathsf{o}_2}{\partial \mathsf{q}_{i_2}} \langle\!\langle q_{i_1}(x_1,t_1), q_{i_2}(x_2,t_2) \rangle\!\rangle^{\mathrm{c}}
$$
$$
+ \frac{1}{2} \sum_{i_1,i_2,i_1',i_2'} \frac{\partial^2 \mathsf{o}_1}{\partial \mathsf{q}_{i_1} \partial \mathsf{q}_{i_1'}} \frac{\partial^2 \mathsf{o}_2}{\partial \mathsf{q}_{i_2} \partial \mathsf{q}_{i_2'}} \langle\!\langle q_{i_1}(x_1,t_1), q_{i_2}(x_2,t_2) \rangle\!\rangle^{\mathrm{c}} \langle\!\langle q_{i_1'}(x_1,t_1), q_{i_2'}(x_2,t_2) \rangle\!\rangle^{\mathrm{c}}
$$
$$
+ \frac{1}{2} \sum_{i_1,i_2,i_1'} \frac{\partial^2 \mathsf{o}_1}{\partial \mathsf{q}_{i_1} \partial \mathsf{q}_{i_1'}} \frac{\partial \mathsf{o}_2}{\partial \mathsf{q}_{i_2}} \frac{1}{2} \sum_{\pm} \langle\!\langle q_{i_1}(x_1 \pm 0^+,t_1), q_{i_1'}(x_1 \mp 0^+,t_1), q_{i_2}(x_2,t_2) \rangle\!\rangle^{\mathrm{c}}
$$
$$
+ \frac{1}{2} \sum_{i_1,i_2,i_2'} \frac{\partial \mathsf{o}_1}{\partial \mathsf{q}_{i_1}} \frac{\partial^2 \mathsf{o}_2}{\partial \mathsf{q}_{i_2} \partial \mathsf{q}_{i_2'}} \frac{1}{2} \sum_{\pm} \langle\!\langle q_{i_1}(x_1,t_1), q_{i_2}(x_2 \pm 0^+,t_2), q_{i_2'}(x_2 \mp 0^+,t_2) \rangle\!\rangle^{\mathrm{c}} + \mathcal{O}(\ell^{-3})
$$

where the first line on the right-hand side is $\mathcal{O}(\ell^{-1})$ and the last three lines are $\mathcal{O}(\ell^{-2})$. In the second line, the point-splitting prescription has been dropped, as it will become clear now that it does not influence the result. Expression (65) simplifies drastically when written for the projected-out observables (63), and this fully accounts for the Drude weight subtraction:

$$
\langle\!\langle (\mathsf{o}_1^-)^{\mathrm{reg}}(\underline{q}(x_1,t_1)), (\mathsf{o}_2^-)^{\mathrm{reg}}(\underline{q}(x_2,t_2)) \rangle\!\rangle^{\mathrm{c}} \tag{66}
$$
$$
= \frac{1}{2} \sum_{i_1,i_2,i_1',i_2'} \frac{\partial^2 \mathsf{o}_1}{\partial \mathsf{q}_{i_1} \partial \mathsf{q}_{i_1'}} \frac{\partial^2 \mathsf{o}_2}{\partial \mathsf{q}_{i_2} \partial \mathsf{q}_{i_2'}} \langle\!\langle q_{i_1}(x_1,t_1), q_{i_2}(x_2,t_2) \rangle\!\rangle^{\mathrm{c}} \langle\!\langle q_{i_1'}(x_1,t_1), q_{i_2'}(x_2,t_2) \rangle\!\rangle^{\mathrm{c}} + \mathcal{O}(\ell^{-3}).
$$

On the right-hand side, we set $x_1 = x, t_1 = t$ and $x_2 = t_2 = 0$, we evaluate

$$
\frac{\partial^2 \mathsf{o}_1}{\partial \mathsf{q}_{i_1} \partial \mathsf{q}_{i_1'}} = \sum_{j_1,j_1'} \langle o_1^-, Q_{j_1}, Q_{j_1'} \rangle^{\mathrm{c}} \mathsf{C}^{j_1,i_1} \mathsf{C}^{j_1',i_1'} \tag{67}
$$

and pass to normal modes using (142), to obtain

$$
\sum_{I_1,I_1',I_2,I_2'} \langle o_1^-, Q_{I_1}, Q_{I_1'} \rangle^{\mathrm{c}} \langle o_2^-, Q_{I_2}, Q_{I_2'} \rangle^{\mathrm{c}} \langle\!\langle q_{I_1}(x,t), q_{I_2}(0,0) \rangle\!\rangle^{\mathrm{c}} \langle\!\langle q_{I_1'}(x,t), q_{I_2'}(0,0) \rangle\!\rangle^{\mathrm{c}}. \tag{68}
$$

This can be evaluated exactly to its leading $\mathcal{O}(\ell^{-2})$ order by using the Euler-scale formula for the two-point functions (152),

$$
\sum_{I,I'} \ell^{-2} \frac{1}{2} \langle o_1^-, Q_I, Q_{I'} \rangle^{\mathrm{c}} \langle o_2^-, Q_I, Q_{I'} \rangle^{\mathrm{c}} \delta(x - \mathsf{v}_I t) \delta(x - \mathsf{v}_{I'} t). \tag{69}
$$

Therefore, taking into account (42) from linear degeneracy, we obtain, in (63),

$$\mathfrak{L}_{o_1,o_2} \overset{\text{micro-micro}}{=} \frac{1}{2}\sum_{I,I'}\int \mathrm{d}t \int \mathrm{d}x\, \langle o_1^-, Q_I, Q_{I'}\rangle^{\mathrm{c}}\langle o_2^-, Q_I, Q_{I'}\rangle^{\mathrm{c}}\delta(x-\mathsf{v}_I t)\delta(x-\mathsf{v}_{I'}t)$$

$$= \frac{1}{2}\sum_{I,I':\mathsf{v}_I\neq\mathsf{v}_{I'}}\int \mathrm{d}t\, \langle o_1^-, Q_I, Q_{I'}\rangle^{\mathrm{c}}\langle o_2^-, Q_I, Q_{I'}\rangle^{\mathrm{c}}\delta((\mathsf{v}_I-\mathsf{v}_{I'})t)$$

$$= \frac{1}{2}\sum_{I,I':\mathsf{v}_I\neq\mathsf{v}_{I'}}\frac{\langle o_1^-, Q_I, Q_{I'}\rangle^{\mathrm{c}}\langle o_2^-, Q_I, Q_{I'}\rangle^{\mathrm{c}}}{|\mathsf{v}_I-\mathsf{v}_{I'}|}. \tag{70}$$

Second, the noise-noise correlation is simply

$$\langle\!\langle \xi_{o_1}(x_1,t_1)\xi_{o_2}(x_2,t_2)\rangle\!\rangle^{\mathrm{c}} = \ell^{-2}\hat{\mathfrak{L}}_{o_1,o_2}\delta(x_1-x_2)\delta(t_1-t_2) \tag{71}$$

so that

$$\mathfrak{L}_{o_1,o_2} \overset{\text{noise-noise}}{=} \hat{\mathfrak{L}}_{o_1,o_2}. \tag{72}$$

Finally, we show that all remaining terms are of order $\mathcal{O}(\ell^{-3})$ or higher orders in $\ell^{-1}$. Let us use for lightness of notation

$$o^{\mathrm{d}}(x,t) = -\frac{\operatorname{sgn}t}{2}\sum_i \mathfrak{D}_o{}^i(\underline{q}(x,t))\partial_x q_i(x,t). \tag{73}$$

The diffusive-diffusive correlation is

$$\langle\!\langle \ell^{-1}o_1^{\mathrm{d}}(x_1,t_1),\ell^{-1}o_2^{\mathrm{d}}(x_2,t_2)\rangle\!\rangle^{\mathrm{c}} = \mathcal{O}(\ell^{-3}) \tag{74}$$

because the two-point function has leading behaviour $\mathcal{O}(\ell^{-1})$, Eq. (20). Similarly, the diffusive-noise correlation is $\langle\!\langle \ell^{-1}o_1^{\mathrm{d}}(x_1,t_1),\ell^{-1}\xi_{o_2}(x_2,t_2)\rangle\!\rangle^{\mathrm{c}} = \mathcal{O}(\ell^{-3})$ because any function of conserved densities evolved in time has a noisy component, coming from integrating the current noise term in the conservation law (26), that is of order $\ell^{-1}$, thus giving the additional factor $\mathcal{O}(\ell^{-1})$.

The diffusive-microcanonical correlation is more subtle. By the cumulant expansion, it is

$$\langle\!\langle o_1^{\mathrm{reg}}(x_1,t_1), o_2^{\mathrm{d}}(x_2,t_2)\rangle\!\rangle^{\mathrm{c}} = -\frac{\operatorname{sgn}t_2}{2\ell}\sum_{i_1,i_2}\frac{\partial\mathsf{o}_1}{\partial\mathsf{q}_{i_1}}\mathfrak{D}_{o_2}{}^{i_2}(\mathsf{q})\partial_{x_2}\langle\!\langle q_{i_1}(x_1,t_1), q_{i_2}(x_2,t_2)\rangle\!\rangle^{\mathrm{c}} + \mathcal{O}(\ell^{-3}) \tag{75}$$

where we have used the fact that $\langle\!\langle \partial_{x_2}q_{i_2}(x_1,t_1)\rangle\!\rangle = 0$ by homogeneity of the state, so that other terms in the cumulant expansion vanish. This is in general of order $\ell^{-2}$. However, it is a total derivative. Because in (61) we integrate over the spatial coordinate, this contribution vanishes in (61). Finally, the microcanonical-noise correlation is, by the cumulant expansion,

$$\langle\!\langle o_1^{\mathrm{reg}}(x_1,t_1), \ell^{-1}\xi_{o_2}(x_2,t_2)\rangle\!\rangle^{\mathrm{c}} = \ell^{-1}\sum_i \frac{\partial\mathsf{o}_1}{\partial\mathsf{q}_i}\langle\!\langle q_i(x_1,t_1), \xi_{o_2}(x_2,t_2)\rangle\!\rangle^{\mathrm{c}} + \mathcal{O}(\ell^{-3}) \tag{76}$$

which again is nonzero at order $\ell^{-2}$, because of the $\mathcal{O}(\ell^{-1})$ noisy component to $q_i(x_1,t_1)$ coming from integrating the hydrodynamic equation. However, under spatial integration the result vanishes:

$$\langle\!\langle Q_i, \xi_{o_2}(x_2,t_2)\rangle\!\rangle^{\mathrm{c}} = 0 \tag{77}$$

because the total charge $Q_i = \int \mathrm{d}x\, q_i(x)$ does not have noisy contribution, as the noise in the hydrodynamic equation appears inside a spatial derivative.

Hence, combining (72) with (70), we obtain

$$\mathfrak{L}_{o_1,o_2} = \frac{1}{2} \sum_{I,I':\mathsf{v}_I \neq \mathsf{v}_{I'}} \frac{\langle o_1^-, Q_I, Q_{I'} \rangle^{\mathrm{c}} \langle o_2^-, Q_I, Q_{I'} \rangle^{\mathrm{c}}}{|\mathsf{v}_I - \mathsf{v}_{I'}|} + \hat{\mathfrak{L}}_{o_1,o_2} \tag{78}$$

from which we deduce the noise strength

$$\hat{\mathfrak{L}}_{o_1,o_2} = \mathfrak{L}_{o_1,o_2} - \frac{1}{2} \sum_{I,I':\mathsf{v}_I \neq \mathsf{v}_{I'}} \frac{\langle o_1^-, Q_I, Q_{I'} \rangle^{\mathrm{c}} \langle o_2^-, Q_I, Q_{I'} \rangle^{\mathrm{c}}}{|\mathsf{v}_I - \mathsf{v}_{I'}|}. \tag{79}$$

This is our main result: in linearly degenerate one-dimensional systems, the noise strength is given by the projected Kubo formula.

## 4.3   Extended Einstein relation

Finally, we establish the relation between the diffusion function $\hat{\mathfrak{D}}_o{}^i(\underline{q})$ in (21) (equiv. (21)), and the noise covariance $\hat{\mathfrak{L}}_{o,j_i}(\underline{q})$ for the observable $o$ and currents $j_i$. For this purpose, let us write, using space-time translation invariance of the state and the conservation laws,

$$
\begin{aligned}
\mathfrak{L}_{o,j_i} &= \ell^2 \int_{-\infty}^{\infty} \mathrm{d}t \int \mathrm{d}x \, \langle\!\langle o^-(x,t), j_i(0,0) \rangle\!\rangle^{\mathrm{c}} \\
&= \ell^2 \int_{-\infty}^{\infty} \mathrm{d}t \int \mathrm{d}x \, \langle\!\langle o^-(0,0), j_i(-x,-t) \rangle\!\rangle^{\mathrm{c}} \\
&= -\ell^2 \int_{-\infty}^{\infty} \mathrm{d}t \int \mathrm{d}x \, x \partial_x \langle\!\langle o^-(0,0), j_i(-x,-t) \rangle\!\rangle^{\mathrm{c}} \\
&= \ell^2 \int_{-\infty}^{\infty} \mathrm{d}t \int \mathrm{d}x \, x \partial_t \langle\!\langle o^-(0,0), q_i(-x,-t) \rangle\!\rangle^{\mathrm{c}} \\
&= \ell^2 \lim_{T\to\infty} \int \mathrm{d}x \, x \langle\!\langle o^-(0,0), \big(q_i(-x,-T) - q_i(-x,T)\big) \rangle\!\rangle^{\mathrm{c}} \\
&= \ell^2 \lim_{T\to\infty} \int \mathrm{d}x \, x \langle\!\langle \big(o^-(x,T) - o^-(x,-T)\big), q_i(0,0) \rangle\!\rangle^{\mathrm{c}}. 
\end{aligned} \tag{80}
$$

We now evaluate this expression by considering in turn the microcanonical, diffusive and noise parts of $o^-(x,T)$ and $o^-(x,-T)$.

For the microcanonical part, the only term that remains from (65) is the third one,

$$\mathfrak{L}_{o,j_i} \overset{\mathrm{micro}}{=} \sum_{\pm'} \pm' \ell^2 \lim_{T\to\infty} \int \mathrm{d}x \, x \frac{1}{2} \sum_{k,k'} \frac{\partial^2 \mathsf{o}}{\partial \mathsf{q}_k \partial \mathsf{q}_{k'}} \frac{1}{2} \sum_{\pm} \langle\!\langle q_k(x \pm 0^+, \pm'T), q_{k'}(x \mp 0^+, \pm'T), q_i(0,0) \rangle\!\rangle^{\mathrm{c}} \tag{81}$$

which we can write, by the inverse of the manipulations (80), as

$$\frac{\ell^2}{4} \int \mathrm{d}t \int \mathrm{d}x \sum_{k,k'} \frac{\partial^2 \mathsf{o}}{\partial \mathsf{q}_k \partial \mathsf{q}_{k'}} \sum_{\pm} \langle\!\langle q_k(x \pm 0^+, t), q_{k'}(x \mp 0^+, t), j_i(0,0) \rangle\!\rangle^{\mathrm{c}}. \tag{82}$$

We now use the nonlinear projection formula for 3-point functions [8, 43]

$$
\langle\!\langle q_k(x \pm 0^+, t), q_{k'}(x \mp 0^+, t), j_i(0,0)\rangle\!\rangle^{\mathrm{c}}
$$

$$
= \sum_l \frac{\partial \mathsf{j}_i}{\partial \mathsf{q}_l} \langle\!\langle q_k(x \pm 0^+, t), q_{k'}(x \mp 0^+, t), q_l(0,0)\rangle\!\rangle^{\mathrm{c}} \tag{83}
$$

$$
+ \sum_{l,l'} \frac{\partial^2 \mathsf{j}_i}{\partial \mathsf{q}_l \partial \mathsf{q}_{l'}} \langle\!\langle q_k(x \pm 0^+, t), q_l(0,0)\rangle\!\rangle^{\mathrm{c}} \langle\!\langle q_{k'}(x \mp 0^+, t), q_{l'}(0,0)\rangle\!\rangle^{\mathrm{c}}. \tag{84}
$$

The first term, when put in (82), gives a total charge $Q_l$ and hence the factor

$$
\frac{\partial}{\partial \beta^l} \langle\!\langle q_k(\pm 0^+, t), q_{k'}(\mp 0^+, t)\rangle\!\rangle^{\mathrm{c}} = 0 \tag{85}
$$

which vanishes by the short-range correlations of the stationary state, and by the point-splitting. The second term is

$$
\frac{\ell^2}{2} \int \mathrm{d}t \int \mathrm{d}x \sum_{k,k'} \frac{\partial^2 \mathsf{o}}{\partial \mathsf{q}_k \partial \mathsf{q}_{k'}} \sum_{l,l'} \frac{\partial^2 \mathsf{j}_i}{\partial \mathsf{q}_l \partial \mathsf{q}_{l'}} \langle\!\langle q_k(x, t), q_l(0,0)\rangle\!\rangle^{\mathrm{c}} \langle\!\langle q_{k'}(x, t), q_{l'}(0,0)\rangle\!\rangle^{\mathrm{c}} \tag{86}
$$

which is exactly the term (66) leading to the result (70),

$$
\mathfrak{L}_{o,j_i} \overset{\mathrm{micro}}{=} \frac{1}{2} \sum_{I,I':\mathsf{v}_I \neq \mathsf{v}_{I'}} \frac{\langle o^-, Q_I, Q_{I'}\rangle^{\mathrm{c}} \langle j_i^-, Q_I, Q_{I'}\rangle^{\mathrm{c}}}{|\mathsf{v}_I - \mathsf{v}_{I'}|}. \tag{87}
$$

For the diffusive part, we use (75)

$$
\mathfrak{L}_{o,j_i} \overset{\mathrm{diffusive}}{=} -\frac{\ell}{2} \sum_{\pm} \pm \lim_{T \to \infty} \int \mathrm{d}x \, x \, \mathrm{sgn}(\pm T) \sum_k \hat{\mathfrak{D}}_o{}^k(\mathsf{q}) \partial_x \langle\!\langle q_k(x, \pm T), q_i(0,0)\rangle\!\rangle^{\mathrm{c}} \tag{88}
$$

giving, by integration by part,

$$
\mathfrak{L}_{o,j_i} \overset{\mathrm{diffusive}}{=} \ell \lim_{T \to \infty} \int \mathrm{d}x \sum_k \hat{\mathfrak{D}}_o{}^k(\mathsf{q}) \langle\!\langle q_k(x, T), q_i(0,0)\rangle\!\rangle^{\mathrm{c}} = \sum_k \hat{\mathfrak{D}}_o{}^k(\mathsf{q}) \mathsf{C}_{ki}. \tag{89}
$$

Finally, the noise part vanishes, $\langle\!\langle \xi_o(x, T), q_i(0,0)\rangle\!\rangle^{\mathrm{c}} = 0$, because $q_i(0,0)$ does not have any noisy component.

Hence we arrive at

$$
\mathfrak{L}_{o,j_i} = \frac{1}{2} \sum_{I,I':\mathsf{v}_I \neq \mathsf{v}_{I'}} \frac{\langle o^-, Q_I, Q_{I'}\rangle^{\mathrm{c}} \langle j_i^-, Q_I, Q_{I'}\rangle^{\mathrm{c}}}{|\mathsf{v}_I - \mathsf{v}_{I'}|} + \sum_k \hat{\mathfrak{D}}_o{}^k(\mathsf{q}) \mathsf{C}_{ki} \tag{90}
$$

and using (44) established above, we find (47).

# 5 Absence of noise in integrable systems

It was recently conjectured that the noise in integrable systems must vanish, see [36, 37, 25]. In particular, in [25] a nonlinear Boltzmann-Gibbs principle in the ballistic $1/\ell$ expansion, as in (21), is used without noise, in order to evaluate diffusive-order corrections to the hydrodynamic equation, showing agreement with numerical simulation. That is, in intregable models, the contribution to the diffusive scale comes solely from the long-range correlations induced by the large-scale variations of the initial state [7], which are of order $1/\ell$ and affect the (point-splitted) microcanonical average.

In this section we provide a proof that this is the case. We show that, under an appropriate choice of the current observables in integrable models – a choice of "gauge" –, which we denote $[j_i]_\infty$, the nonlinear fluctuating Boltzmann-Gibbs principle simplifies to a non-fluctuating one at all orders of the hydrodynamic expansion:

$$\overline{[j_i]_\infty}(x,t) \text{ in } \langle\cdots\rangle_\ell \longrightarrow j_i(x,t) := \mathrm{j}_i^{\mathrm{reg}}(\underline{q}(x,t)) + \mathcal{O}(\ell^{-\infty}) \text{ in } \langle\!\langle\cdots\rangle\!\rangle_\ell. \tag{91}$$

At the diffusive order, with $\mathcal{O}(\ell^{-2})$ instead of $\mathcal{O}(\ell^{-\infty})$, the result holds for the PT-symmetric current $j_i$ instead of $[j_i]_\infty$ (that is, without the need to further fix the gauge).

$$\overline{j_i}(x,t) \text{ in } \langle\cdots\rangle_\ell \longrightarrow j_i(x,t) := \mathrm{j}_i^{\mathrm{reg}}(\underline{q}(x,t)) + \mathcal{O}(\ell^{-2}) \text{ in } \langle\!\langle\cdots\rangle\!\rangle_\ell. \tag{92}$$

First, we show the result at the diffusive order (92), with the PT-symmetric current $j_i$. This is done using previous, independent results on the Onsager matrix. We show that in integrable systems, the noise $\xi_{j_i}$ for the currents $j_i$, as appears at order $\ell^{-1}$ in (21), vanishes, and as a consequence, the bare diffusion for all observables also vanishes. That is,

$$\xi_{j_i} = 0 \tag{93}$$

and more specifically

$$\hat{\mathfrak{L}}_{j_i,j_i}(\underline{\mathsf{q}}) = 0 \;\Rightarrow\; \hat{\mathfrak{L}}_{o,j_i}(\underline{\mathsf{q}}) = \hat{\mathfrak{L}}_{j_i,o}(\underline{\mathsf{q}}) = 0 \;\Rightarrow\; \hat{\mathfrak{D}}_o^{\;k}(\underline{\mathsf{q}}) = 0 \tag{94}$$

where the last equality follows from (47).

Second, we give a first-principle argument, based on the fluid-cell relaxation picture of Sec. 3.2. This argument is important in two ways:

(1) It provides an *independent, first-principle derivation of the result for the Onsager matrix in integrable models*, Eq. (95). Indeed, as we have shown that the noise strength is the projected Kubo formula, a corollary that this vanishes is that the un-projected, full Kubo formula is given by its projection onto quadratic charges. This reproduces the known expression for the Onsager matrix in integrable systems in terms of quasi-particles.

(2) It allows us to show that in fact, the noise for currents must vanish *at all orders in the hydrodynamic expansion*, Eq. (91). This is true, beyond the diffusive scale, as long as current observables are chosen appropriately. Recall that charge densities and currents are defined only up to total derivatives of local observables – this is the "gauge" freedom in determining densities and currents. At the diffusive level, this gauge freedom is lifted by choosing the PT gauge [41]. But beyond it, there is no natural gauge, see however a discussion in [64]. Our proof

provides a partially constructive way of choosing the gauge, that guarantees that the nonlinear Boltzmann-Gibbs principle for currents does not admit noise at all orders of the hydrodynamic expansion.

*First proof.* We show that the absence of noise at the diffusive scale, Eq. (92) or equivalently (94), is a consequence of our main result (44), along with the following observation.

In the hard rods model, the Onsager matrix $\mathfrak{L}_{j_i,j_k}$, which is in fact an integral operator as it is taken in the basis of the conserved hard rods velocity $i = v \in \mathbb{R}$, was evaluated exactly, see e.g. [45, 5, 65]. Further, in [40, 41] the Onsager matrix $\mathfrak{L}_{j_i,j_k}$ was evaluated in integrable models using quantum form factor techniques, generalising the hard-rod formula (see also [42] where the diagonal part was evaluated using a quasi-particle picture). Then, it was shown in [24] that this general formula in fact corresponds to a projection, within the diffusive Hilbert space $\mathcal{H}_{\mathrm{dif}}$, onto the scatteing space formed by quadratic charges (see the discussion in Sec. 4)

$$\mathfrak{L}_{j_i,j_k} \overset{\text{integrable models}}{=} \left(\mathbb{P}_{\mathrm{scat}} j_i, \mathbb{P}_{\mathrm{scat}} j_k\right)_{\mathrm{dif}} = \frac{1}{2} \sum_{I \neq I'} \frac{\langle j_i^-, Q_I, Q_{I'} \rangle^{\mathrm{c}} \langle j_k^-, Q_I, Q_{I'} \rangle^{\mathrm{c}}}{|\mathsf{v}_I - \mathsf{v}_{I'}|}. \tag{95}$$

Therefore, from these results and (44), we conclude that the noise, at least at the leading order, has vanishing correlations

$$\hat{\mathfrak{L}}_{j_i,j_k} \overset{\text{integrable models}}{=} 0. \tag{96}$$

By the Cauchy-Schwartz inequality (see [24] for the inner-product structure of the Kubo formula), we have

$$(\hat{\mathfrak{L}}_{o,j_i})^2 \leq \hat{\mathfrak{L}}_{o,o} \hat{\mathfrak{L}}_{j_i,j_i} \overset{\text{integrable models}}{=} 0 \tag{97}$$

which shows (94).

*Second proof.* We show (92), and then more generally (91), using a general first-principle argument based on relaxation of fluid-cell means and fundamental properties of integrable systems.

In many-body integrable models, there exists an infinite number of commuting, independent "higher" time flows, besides the "usual" time evolution. If the integrable system is Hamitonian, then these are generated by infinitely many linearly and functionally independent commuting, extensive Hamiltonians. In translation-invariant systems, these time flows are also translation invariant. That is, we can evolve any local observable not just in time $\mathsf{t} = \mathsf{t}_1$, but also in times $\mathsf{t}_2$, $\mathsf{t}_3$, ..., which we denote by the commuting one-parameter groups of maps $\tau_2^{\mathsf{t}_2}, \tau_3^{\mathsf{t}_3}, \ldots$ acting on observables:

$$o(\mathsf{x}, \mathsf{t}_1, \mathsf{t}_2, \mathsf{t}_3, \ldots) = \tau_2^{\mathsf{t}_2} \tau_3^{\mathsf{t}_3} \cdots o(\mathsf{x}, \mathsf{t}_1), \quad \mathsf{t}_1, \mathsf{t}_2, \mathsf{t}_3, \ldots \in \mathbb{R}. \tag{98}$$

Crucially, because they are commuting flows, each conserved density $q_i$ is also a conserved density with respect to all higher times $t_k$, with associated currents $j_{ki}$:

$$\partial_{\mathsf{t}_k} q_i(\mathsf{x}, \mathsf{t}_1, \ldots, t_n) + \partial_{\mathsf{x}} j_{ki}(\mathsf{x}, \mathsf{t}_1, \ldots, \mathsf{t}_n) = 0. \tag{99}$$

Consistency requires $\partial_{\mathsf{t}_k} j_{li} = \partial_{\mathsf{t}_l} j_{ki}$. Further, using this, it was shown in [24, App 4] that the Onsager matrix is *invariant* under these flows,

$$\mathfrak{L}_{\tau_2^{\mathsf{t}_2} \tau_3^{\mathsf{t}_3} \cdots j_i, o} = \mathfrak{L}_{j_i, o}. \tag{100}$$

It is simple to see that the same holds for the Onsager matrix from which quadratic charges have been projected out, as these are invariant under all higher time flows,

$$\hat{\mathfrak{L}}_{\tau_2^{t_2}\tau_3^{t_3}\cdots j_i, o} = \hat{\mathfrak{L}}_{j_i, o}. \tag{101}$$

Now recall the argument in Sec. 3.2 for the origin of hydrodynamic noise. On a fluid cell of size $L$, after a time $T$, there has been $\mathcal{O}(LT) = \mathcal{O}(\ell)$ fluctuation events for the fluid-cell mean $\overline{o}$ of the observable $o$, while, thanks to the conservation laws, the conserved densities $\overline{q_i}$'s are essentially constant; see Fig. 2. Hence, taking an average over a time $t \in [0, T]$ (or, equivalently, not taking the time average, but using typicality) relaxation to the microcanonical average occurs, plus a noise of variance $\propto 1/\ell$ by the central limit theorem.

Consider $o = j_i$ to be a current. Let us change this observable by averaging over $n$ of the higher time flows, with higher times in the range $[0, T']$ for some fixed $T' > 1$ (not scaling with $\ell$),

$$[j_i] = \frac{1}{T'^n} \int_0^{T'} \mathrm{dt}_2 \cdots \int_0^{T'} \mathrm{dt}_{n+1} \prod_{k=2}^{n+1} \tau_k^{t_k} j_i. \tag{102}$$

Taking the time-average over $t_1 \in [0, T]$, the fluid-cell mean $\overline{[j_i]}$ is the average of the original fluid cell mean $\overline{j_i}$ over the time-hyperbox $[0, T] \times [0, T']^n$ of dimension $n+1$, with $t_1 \in [0, T]$ and $t_k \in [0, T']$ for $k = 2, \ldots, n+1$. On this hyperbox, fluid cell means of conserved densities $\overline{q_i}$'s are all essentially constant, because of (99). As each time flow is independent, the fluctuations induced, from typical configurations, by each flow are also independent. Thus, on the space-time hyperbox $[0, L] \times [0, T] \times [0, T']^n$ the configuration shows a number of independent fluctuation events (particle collisions, etc.). In the higher-dimensional equivalent of Fig. 2, these fluctuation events lie on hypersurfaces of dimension $n + 1$ criss-crossing the hyperbox. There are $\mathcal{O}(LT)$ such hypersurfaces, each of hypervolume $\propto T'^n$. It is natural to assume that on each such hypervolume, there is typically $T'^n/\ell_{\mathrm{micro}}^n$ independent changes, by an extension of the assumption of molecular chaos to this context with higher-dimensional time. Therefore, the fluid cell mean $\overline{[j_i]}$ has undergone $\propto T'^n$ times more fluctuations than $\overline{j_i}$.

Taking $n$ and/or $T'$ large after the $\ell \to \infty$ asymptotic, we thus find by the central limit theorem that the noise for the observable $[j_i]$ has variance

$$\langle\!\langle \xi_{[j_i]}(x, t) \xi_{[j_i]}(x', t') \rangle\!\rangle_\ell \sim c(T')\ell^{-1}\delta(x - x')\delta(t - t'), \quad c(T') = \mathcal{O}(T'^{-n}) \tag{103}$$

hence

$$\hat{\mathfrak{L}}_{[j_i],[j_i]} = \mathcal{O}(T'^{-n}). \tag{104}$$

By (101), this means

$$\hat{\mathfrak{L}}_{j_i, j_i} = \mathcal{O}(T'^{-n}). \tag{105}$$

As $n$ and $T' > 0$ can be taken as large as we wish, this implies $\hat{\mathfrak{L}}_{j_i, j_i} = 0$, and by Cauchy Schwartz shows (94).

We note that the presence of *a single* higher time flow, $n = 1$, is sufficient for this argument, as we can take $T' > 1$ as large as we wish (but according to integrability theory, a single higher-flow implies the presence of infinitely-many higher flows).

Now let us consider the observable $[j_i]$. Because the higher flows commute, this is a good current observable, for a conserved density that is likewise averaged over higher time flows:

$$\partial_{\mathrm{x}}[q_i] + \partial_{\mathrm{t}}[j_i] = 0. \tag{106}$$

The above argument shows that as we make $n$ larger for any fixed $T' > 1$, the emergent noise's variance becomes smaller. By the Bienaymé-Chebyshev theorem, this implies that all higher cumulants are likewise made smaller. Therefore, we find that the following particular choice of densities and currents,

$$[q_i]_\infty = \lim_{n \to \infty} [q_i], \quad [j_i]_\infty = \lim_{n \to \infty} [j_i] \tag{107}$$

are good choices of gauge that guarantee that *the associated current in the fluctuating hydrodynamic theory has vanishing noise at all orders of the hydrodynamic expansion*, and we obtain (91).

**Remark 5.1.** *Note how this is related to the natural conjecture that semiclassical Bethe particles [66, 67, 68] give a good representation of generalised hydrodynamics at all orders [68, 69], although still missing is the connection between the above choice of gauge and the choice that associates currents to these particles' flows.*

## 6   Hydrodynamic equation and diffusive two-point functions

Sec. 4.2 and 4.3 illustrate some of the techniques involved in calculating using the nonlinear fluctuating Boltzmann-Gibbs principle (21), in particular combining the point-splitting regularisation with the cumulant expansion. In order to further illustrate this "calculus" and to show the general consistency of our framework, in this section we provide two simple applications. We obtain the hydrodynamic equation up to the diffusive correction, first for one-point function in non-equilibrium states emanating from (12), and second for two-point functions of conserved densities in stationary states.

For one-point function, the result is highly anomalous, taking the form of a coupled system of equations as in [25] for integrable systems, but generalised to the presence of bare diffusion. The calculation is a simple generalisation of that shown in the main text of [25].

For two-point functions, the result is a normal diffusion equation, with the full diffusion matrix related by Einstein relation to the *full, un-projected Onsager matrix*. This is usually assumed to be valid and follows from sum-rule manipulations, see e.g. [5, 41]. Here we derive it from our fluctuating hydrodynamic theory, in which context it is somewhat non-trivial, as the effects of long-range correlations to the diffusive scale – coming from the regularised microcanonical part of (21) – and of bare diffusion must combine just in the right way to give the diffusion matrix related to the full Onsager matrix. This was also derived in [25] for integrable systems, although in this case there is no bare diffusion. It was derived by a different technique, using a special linear response theory on the hydrodynamic equation for one-point functions in non-equilibriun states. There, the full Onsager matrix – without bare diffusion – arises because the at infinitesimal macroscopic times, the long-range correlations exactly reproduce the projection onto quadratic charges, thus, for integrable systems the full Onsager matrix (as we showed in

Sec. 5). We instead perform a direct calculation using the nonlinear fluctuating Boltzmann-Gibbs principle, which involves 3-point functions in stationary states recently evaluated [43].

Thus we show that the present framework gives a consistent theoretical underpinning for both the special emergent hyrodynamic equation out of equilibrium, and normal diffusion at the linearised level.

## 6.1 The hydrodynamic equations at the diffusive scale

Using our fluctuating hydrodynamic theory based on (21), we first write down the diffusive-scale hydrodynamic equation, taking into account the results Points I-III. For definiteness the initial state is the long-wavelength state (12), but more general states would lead to the same equation.

The hydrodynamic equation is obtained from the conservation laws:

$$\partial_t \langle\!\langle q_i(x,t) \rangle\!\rangle_\ell + \partial_x \langle\!\langle j_i(x,t) \rangle\!\rangle_\ell = 0. \tag{108}$$

Inserting (21) for the case $o = j_i$ in this, using $\langle\!\langle \xi_o(x,t) \rangle\!\rangle_\ell = 0$ and the fact that for the diffusive order $\ell^{-1}$ we may use the leading cumulant-expansion form

$$\sum_i \langle\!\langle \hat{\mathfrak{D}}_o{}^i(\underline{q}(x,t)) \partial_x q_i(x,t) \rangle\!\rangle = \sum_i \hat{\mathfrak{D}}_o{}^i(\langle\!\langle \underline{q}(x,t) \rangle\!\rangle_\ell) \partial_x \langle\!\langle q_i(x,t) \rangle\!\rangle_\ell + \mathcal{O}(\ell^{-1}) \tag{109}$$

and denoting for ligthness $\mathsf{q}_i(x,t) = \langle\!\langle q_i(x,t) \rangle\!\rangle_\ell$, this gives

$$\partial_t \mathsf{q}_i(x,t) + \partial_x \langle\!\langle \mathsf{j}_i^{\mathrm{reg}}(\underline{q}(x,t)) \rangle\!\rangle_\ell = \frac{1}{2\ell} \sum_k \partial_x \Big( \hat{\mathfrak{D}}_i{}^k(\underline{\mathsf{q}}(x,t)) \partial_x \mathsf{q}_k(x,t) \Big) \tag{110}$$

where we used the notation (27). The quantity $\langle\!\langle \mathsf{j}_i^{\mathrm{reg}}(\underline{q}(x,t)) \rangle\!\rangle_\ell$ was evaluated in [25], and depends on the long-range correlations generated by the long-wavelength state [7], which are at the diffusive order $\ell^{-1}$. These long-range correlations themselves satisfy a set of Euler-scale evolution equation. The result is a set of two combined equations:

$$\partial_t \mathsf{q}_i(x,t) + \sum_k \mathsf{A}_i{}^k(x,t) \partial_x \mathsf{q}_k(x,t) + \frac{1}{4\ell} \sum_{I,J,\pm} \partial_x \Big( \langle j_i^-, Q_I, Q_J \rangle^{\mathrm{c}}_{\underline{\beta}(x,t)} E_{IJ}(x \pm 0^+, x \mp 0^+; t) \Big)$$
$$= \frac{1}{2\ell} \sum_k \partial_x \Big( \hat{\mathfrak{D}}_i{}^k(\underline{\mathsf{q}}(x,t)) \, \partial_x \mathsf{q}_k(x,t) \Big) \tag{111}$$

with

$$\partial_t E_{IJ}(x,y;t) + \partial_x \big( \mathsf{v}_I(x,t) E_{IJ}(x,y;t) \big) + \partial_y \big( \mathsf{v}_J(y,t) E_{IJ}(x,y;t) \big)$$
$$= \delta(x-y) \sum_k \langle j_k^-, Q_I, Q_J \rangle^{\mathrm{c}}_{\underline{\beta}(x,t)} \partial_x \beta^k(x,t). \tag{112}$$

Here $\beta^i(x,t) := \beta^i(\mathsf{q}(x,t))$ (see Eq. (11)) and likewise for the hydrodynamic velocities $\mathsf{v}_I(x,t)$. In the state (12), the initial conditions are

$$\beta^i(x,0) = \beta^i(x), \quad E_{rs}(x,y;0) = 0 \tag{113}$$

but these equations are more general, and valid also for initial states that admit non-trivial long-range correlations $E_{rs}(x, y; 0) \neq 0$. Eqs. (111), (112) generalise [25, Eqs 9, 10], which was for integrable models, to the inclusion of the bare diffusion generically present away from integrability.

For integrable models, Eqs. (111) and (112) are immediate to write in terms of quasi-particle rapidities, using that, in normal modes $I \in \mathbb{R}$ (which are of rapidity type), $\mathsf{A}_I{}^J = \delta(I - J)v^{\text{eff}}(I)$, and the three-point coupling in normal modes (143) has the expression displayed in [43, Eq 103]. See [69] where it is written in even more generality, including external force terms [70].

## 6.2 Normal diffusion for stationary-state two-point functions

We restrict to stationary states

$$\langle \cdots \rangle = \langle \cdots \rangle_{\underline{\beta}} \tag{114}$$

and all constitutive functions are evaluated within this state (we keep their state dependence implicit).

We will show that the diffusive equation for two-point functions of conserved densities takes a diffusive form:

$$\partial_t \langle \overline{q_i}(x,t), \overline{q_j}(0,0) \rangle^{\text{c}} + \sum_k \mathsf{A}_i{}^k \partial_x \langle \overline{q_k}(x,t), \overline{q_j}(0,0) \rangle^{\text{c}} = \frac{1}{2\ell} \sum_k \mathfrak{D}_i{}^k \partial_x^2 \langle \overline{q_k}(x,t), \overline{q_j}(0,0) \rangle^{\text{c}}, \quad t > 0. \tag{115}$$

This is in principle valid for fluid-cell means as written here, so that possible oscillatory behaviours are averaged out, but in practice often the fluid-cell mean is not required. Importantly, as mentioned, the diffusion matrix involved is related by the Einstein relation to the *full* Onsager matrix,

$$\mathfrak{L}_{il} = \sum_k \mathfrak{D}_i{}^k \mathsf{C}_{kl}. \tag{116}$$

Here and below we use the simplified notation $\mathfrak{L}_{ik} = \mathfrak{L}_{j_i, j_k}$ paralleling (27). Eq. (116) is the same relation as Eq. (47) for the bare diffusion matrix in terms of the projected Kubo formula.

That the full diffusion matrix be involved in the two-point function is non-trivial: the bare diffusion term easily gives a contribution to the diffusion term in (115), however the long-range correlations, coming from the regularised microcanonical average, also give a contribution, and these sum up exactly to the full diffusion matrix. This is what we will show now. The proof is based on a result for the exact three-point function of conserved densities at the Euler scale that we obtained recently by projection techniques, see [43, Sec 5.2].

Using the conservation laws again, passing to the fluctuating theory, and using normal modes and the fact that two-point functions of conserved densities at the Euler scale are diagonal in this basis, Eq. (152), we must show that

$$\langle\!\langle j_I(x,t), q_J(0,0) \rangle\!\rangle^{\text{c}} = \delta_{IJ} \mathsf{v}_I \langle\!\langle q_I(x,t), q_I(0,0) \rangle\!\rangle^{\text{c}} - \frac{1}{2\ell} \mathfrak{L}_{IJ} \partial_x \langle\!\langle q_J(x,t), q_J(0,0) \rangle\!\rangle^{\text{c}}, \quad t > 0. \tag{117}$$

As the second term on the right-hand side has a factor $1/\ell$, it can be evaluated at the Euler scale, so we must show

$$\langle\!\langle j_I(x,t), q_J(0,0) \rangle\!\rangle^{\text{c}} = \delta_{IJ} \mathsf{v}_I \langle\!\langle q_I(x,t), q_I(0,0) \rangle\!\rangle^{\text{c}} - \frac{1}{2\ell^2} \mathfrak{L}_{IJ} \delta'(x - \mathsf{v}_J t), \quad t > 0. \tag{118}$$

*Proof.* We use (21) for $o = j_I$. Using (32), as the noise $\xi_I$ does not correlate with the fluctuation of $q_J(0,0)$, induced by the initial state, it does not contribute:

$$\langle\!\langle j_I(x,t), q_J(0,0)\rangle\!\rangle^{\text{c}} \overset{\text{noise}}{=} 0. \tag{119}$$

The diffusive term can be evaluated by the cumulant expansion, at leading order giving simply

$$\langle\!\langle j_I(x,t), q_J(0,0)\rangle\!\rangle^{\text{c}} \overset{\text{diffusive}}{=} -\frac{1}{2\ell}\hat{\mathfrak{L}}_{IJ}\partial_x\langle\!\langle q_J(x,t), q_J(0,0)\rangle\!\rangle^{\text{c}} = -\frac{1}{2\ell^2}\hat{\mathfrak{L}}_{IJ}\delta(x - \mathsf{v}_J t), \quad t > 0. \tag{120}$$

This is the bare contribution to the diffusive term of the two-point function.

The most important part is that coming from the microcanonical average, with point-splitting regularisation. Using (65), this is

$$\langle\!\langle \mathsf{j}_I^{\text{reg}}(\underline{q}(x,t))q_J(0,0)\rangle\!\rangle^{\text{c}} \tag{121}$$
$$= \delta_{IJ}\mathsf{v}_I\langle\!\langle q_I(x,t), q_J(0,0)\rangle\!\rangle^{\text{c}} + \frac{1}{2}\sum_{KL}\mathsf{A}_I^{KL}\frac{1}{2}\sum_{\epsilon=\pm0^+}\langle\!\langle q_K(x+\epsilon,t), q_L(x-\epsilon,t), q_J(0,0)\rangle\!\rangle^{\text{c}}.$$

We must now evaluate the three-point function. For this purpose, it is simpler to bring the space-time variables $x, t$ onto $q_J$ by translation invariance and PT symmetry:

$$\langle\!\langle q_K(x+\epsilon,t), q_L(x-\epsilon,t), q_J(0,0)\rangle\!\rangle^{\text{c}} = \langle\!\langle q_K(-\epsilon,0), q_L(+\epsilon,0), q_J(x,t)\rangle\!\rangle^{\text{c}}. \tag{122}$$

We then use the formula [43, Eqs 98, 99]. This has two types of contribution:

$$\langle\!\langle q_K(-\epsilon,0), q_L(\epsilon,0), q_J(x,t)\rangle\!\rangle^{\text{c}} = \ell^{-2}(C_{\text{linear}} - C_{\text{nonlinear}}). \tag{123}$$

The first is a direct, linear propagation. This contribution vanishes because of the point-splitting:

$$C_{\text{linear}} = \langle Q_K, Q_L, q_J\rangle^{\text{c}}\delta(x + \epsilon - \mathsf{v}_J t)\delta(x - \epsilon - \mathsf{v}_J t) = 0. \tag{124}$$

The second is an indirect, nonlinear propagation, coming from the nonlinear part of the 3-point projection. Different cases in principle must be treated, depending if some of the velocities $\mathsf{v}_J$, $\mathsf{v}_K$, $\mathsf{v}_L$ are equal to each other. However, the expression in the case where only two velocities are equal to each other, can be obtained by taking the formal limit of equal velocities; while that where all velocities are equal to each other does not contribute because of the vanishing 3-point coupling (161). Hence, it is sufficient to perform the full calculation from the expression for unequal velocities. Because of our choice of values of times, only the nonlinear propagation associated to $q_J$ is required, giving:

$$C_{\text{nonlinear}} = \mathsf{A}_J^{KL}\delta'(v'u - vu')\big(|v'|(s(u' + v't) - s(u')) - |v|(s(u + vt) - s(u))\big) = \mathsf{A}_J^{KL}c \tag{125}$$

where $s(a) = \frac{1}{2}\text{sgn}(a)$ and

$$u = x - \mathsf{v}_J t + \epsilon, \ u' = x - \mathsf{v}_J t - \epsilon, \ v = \mathsf{v}_{JK}, \ v' = \mathsf{v}_{JL} \tag{126}$$

with the assumption $\mathsf{v}_J \neq \mathsf{v}_K$, $\mathsf{v}_J \neq \mathsf{v}_L$, $\mathsf{v}_K \neq \mathsf{v}_L$. Thus we must evaluate:

$$c = \delta'\big(\mathsf{v}_{KL}(x - \mathsf{v}_J t) + \epsilon(2\mathsf{v}_J - \mathsf{v}_K - \mathsf{v}_L)\big) \times \tag{127}$$
$$\times \Big(|\mathsf{v}_{JL}|(s(x - \mathsf{v}_L t - \epsilon) - s(x - \mathsf{v}_J t - \epsilon)) - |\mathsf{v}_{JK}|(s(x - \mathsf{v}_K t + \epsilon) - s(x - \mathsf{v}_J t + \epsilon))\Big)$$

We evaluate the term with $s(x - \mathsf{v}_J t - \epsilon)$ as

$$-\frac{|\mathsf{v}_{JL}|}{\mathsf{v}_{KL}|\mathsf{v}_{KL}|}\delta'\left(x - \mathsf{v}_J t + \epsilon\frac{2\mathsf{v}_J - \mathsf{v}_K - \mathsf{v}_L}{\mathsf{v}_{KL}}\right)s(x - \mathsf{v}_J t - \epsilon)$$

$$\overset{|\epsilon|\to 0}{=} \frac{|\mathsf{v}_{JL}|}{\mathsf{v}_{KL}|\mathsf{v}_{KL}|}\left(\delta'(x - \mathsf{v}_J t)s\left(\frac{\mathsf{v}_{JL}}{\mathsf{v}_{KL}}\right)\mathrm{sgn}(\epsilon) + \delta(x - \mathsf{v}_J t)\delta\left(\frac{\mathsf{v}_{JL}}{\mathsf{v}_{KL}}\epsilon\right)\right)$$

$$= \frac{|\mathsf{v}_{JL}|}{\mathsf{v}_{KL}|\mathsf{v}_{KL}|}\delta'(x - \mathsf{v}_J t)s\left(\frac{\mathsf{v}_{JL}}{\mathsf{v}_{KL}}\right)\mathrm{sgn}(\epsilon) \tag{128}$$

and under the sum $\sum_{\epsilon=\pm 0^+}$ in (121) the result vanishes. Likewise for $s(x - \mathsf{v}_J t + \epsilon)$,

$$\frac{|\mathsf{v}_{JK}|}{\mathsf{v}_{KL}|\mathsf{v}_{KL}|}\delta'\left(x - \mathsf{v}_J t + \epsilon\frac{2\mathsf{v}_J - \mathsf{v}_K - \mathsf{v}_L}{\mathsf{v}_{KL}}\right)s(x - \mathsf{v}_J t + \epsilon) \overset{|\epsilon|\to 0}{=} -\frac{|\mathsf{v}_{JK}|}{\mathsf{v}_{KL}|\mathsf{v}_{KL}|}\delta'(x - \mathsf{v}_J t)s\left(\frac{\mathsf{v}_{JK}}{\mathsf{v}_{KL}}\right)\mathrm{sgn}(\epsilon) \tag{129}$$

which again vanishes under $\sum_{\epsilon=\pm 0^+}$. For the term with $s(x - \mathsf{v}_L t - \epsilon)$ we get

$$\frac{|\mathsf{v}_{JL}|}{\mathsf{v}_{KL}|\mathsf{v}_{KL}|}\delta'\left(x - \mathsf{v}_J t + \epsilon\frac{2\mathsf{v}_J - \mathsf{v}_K - \mathsf{v}_L}{\mathsf{v}_{KL}}\right)s(x - \mathsf{v}_L t - \epsilon)$$

$$\overset{|\epsilon|\to 0}{=} \frac{|\mathsf{v}_{JL}|}{\mathsf{v}_{KL}|\mathsf{v}_{KL}|}\left(\delta'(x - \mathsf{v}_J t)s(\mathsf{v}_{JL}t) - \delta(x - \mathsf{v}_J t)\delta(\mathsf{v}_{JL}t)\right)$$

$$\overset{t\geq 0}{=} \frac{\mathsf{v}_{JL}}{2\mathsf{v}_{KL}|\mathsf{v}_{KL}|}\delta'(x - \mathsf{v}_J t) \tag{130}$$

and likewise for the term with $s(x - \mathsf{v}_K t + \epsilon)$,

$$-\frac{|\mathsf{v}_{JK}|}{\mathsf{v}_{KL}|\mathsf{v}_{KL}|}\delta'\left(x - \mathsf{v}_J t + \epsilon\frac{2\mathsf{v}_J - \mathsf{v}_K - \mathsf{v}_L}{\mathsf{v}_{KL}}\right)s(x - \mathsf{v}_K t + \epsilon) \overset{|\epsilon|\to 0,\, t>0}{=} -\frac{\mathsf{v}_{JK}}{2\mathsf{v}_{KL}|\mathsf{v}_{KL}|}\delta'(x - \mathsf{v}_J t). \tag{131}$$

Combining,

$$\frac{1}{2}\sum_{\epsilon=\pm 0^+}C_{\mathrm{nonlinear}} = \mathsf{A}_J^{KL}\frac{1}{2|\mathsf{v}_{KL}|}\delta'(x - \mathsf{v}_J t) \tag{132}$$

and therefore

$$\frac{1}{2}\sum_{KL}\mathsf{A}_I^{KL}\frac{1}{2}\sum_{\epsilon=\pm 0^+}\langle\!\langle q_K(x+\epsilon,t), q_L(x-\epsilon,t), q_J(0,0)\rangle\!\rangle^{\mathrm{c}} \overset{t\geq 0}{=} -\sum_{KL}\frac{\mathsf{A}_I^{KL}\mathsf{A}_J^{KL}}{4|\mathsf{v}_{KL}|\ell^2}\delta'(x - \mathsf{v}_J t). \tag{133}$$

With (121), (120) and (133), we therefore find

$$\langle\!\langle j_I(x,t), q_J(0,0)\rangle\!\rangle^{\mathrm{c}} = \delta_{IJ}\mathsf{v}_I\langle\!\langle q_I(x,t), q_J(0,0)\rangle\!\rangle^{\mathrm{c}} - \frac{1}{2\ell^2}\left(\hat{\mathfrak{L}}_{IJ} + \sum_{KL}\frac{\mathsf{A}_I^{KL}\mathsf{A}_J^{KL}}{2|\mathsf{v}_{KL}|}\right)\delta'(x - \mathsf{v}_J t). \tag{134}$$

Thus from (44) and (143), we have shown that (118) holds. ∎

# 7  Conclusion

We have developed a hydrodynamic fluctuation theory for linearly degenerate systems in one spatial dimension, up to, including, the diffusive order $1/\ell$ as $\ell \to \infty$ in the ballistic scaling of space and time, $\mathrm{x} = \ell x$, $\mathrm{t} = \ell t$. The theory is presented as a stochastic PDE in a $1/\ell$ expansion, which is well defined up to the first subleading order. This is based on a nonlinear fluctuating Boltzmann-Gibbs principle, where generic observables are projected onto conserved densities, with the addition of diffusive and noise terms. This nonlinear fluctuating Boltzmann-Gibbs principle is justified by heuristic arguments that we have disccused carefully, based on the separation of relaxation time-scales between coarse-grained conserved densities and other coarse-grained observables. Relaxation of arbitrary observables to their microcanonical averages is modified by noise via the central limit theorem, by "bare" diffusion via because of relaxation time delay (fluctuation-dissipation), and by a non-fluctuating, non-diffusive correction to the microcanonical average, representing the correction to the microcanonical average due to the finite size of the region (the fluid-cell) on which it is taken. The noise covariance is shown to be given by a projected Kubo formula, where diffusive-scale effects of long-range correlations [25] are subtracted (projected out, from a Hilbert space perspective [24]). The diffusion is shown to be obtained by Einstein relation from this, and the microcanonical correction is shown to give a point-splitting regularisation [25] that makes the stochastic PDE, with delta-correlations both in noise and initial conditions, well-defined. We show, in two different ways, that the noise vanishes at the diffusive scale in integrable models, proving conjectures made previously [36, 37, 25]. We show that, under an appropriate choice of currents, their noise in fact vanishes at all orders in the hydrodynamic expansion. We write down the full hydrodynamic equation for linearly degenerate systems, including effects of long-range correlations [25] and bare diffusion. We also derived, within our fluctuating hydrodynamic formalism, the diffusive hydrodynamic equation for dynamical correlation functions in stationary states, where we showed that diffusion is associated to the full Onsager matrix instead of the projected one, as it should be.

In the heuristic arguments we have provided to explain the form of the nonlinear Boltzmann-Gibbs principle in Sec. 3.2, noise emerges under the usual assumption of molecular chaos, here expressed as the assumption that interaction events are essentially independent and non-correlated when separated enough in space and time (see Fig. 2). Clearly, as we show that noise in fact vanishes for currents in integrable models, there may in fact be correlations that reduce the noise even for certain observables which are not conserved densities. In Hamiltonian systems, one of the leading open problems in the hydrodynamic theory remains to more accurately understand the emergence of Gaussian white noise, perhaps by expanding on the heuristic arguments we have expressed.

In order to make the absence of noise for currents in integrable models more explicit, and to put the heuristic arguments of Sec. 3.2 on a more solid basis, one should investigate numerically fluctuations of observables on fluid cells[6].

It has recently been shown with mathematical rigour that in (generalised) hard rods models, a certain fluctuating hydrodynamic equation emerges for linear fluctuating fields, where noise appears, albeit with strong correlations in space [71]. This noise is *not* of the type that appears

---

[6]As we mentioned, shortly after the first version of this paper was posted, paper [46] performed this in the hard rods model, confirming our results.

in (21), which, as we have shown, vanishes in integrable models. Instead, we believe this space-correlated noise is a "remnant", at the linear level, of long-range correlations that are known to occur from the present nonlinear version of the fluctuating hydrodynamics. More investigation on this aspect would be appropriate.

It would be interesting to evaluate explicitly, using the present theory, the first subleading corrections to correlation functions and to large-deviation functions beyond the Euler scale [7, 8, 43]. It would also be interesting to generalise this to higher dimensions (see [43] for the nonlinear projection principles for correlation functions at the leading Euler scale), where long-range correlations do not affect the diffusive scale but may affect higher scales of the hydrodynamic expansion.

Finally, going beyond linearly degenerate systems would be very interesting. It is possible that a regularisation similar to that which we introduced in order to make the fluctuating hydrodynamic theory well defined in its asymptotic expansion in $1/\ell$, be related to the regularisation necessary to make the KPZ equation well defined [72]. We also propose that bare diffusion and hydrodynamic noise are given by similar projection principles even more generally, in one-dimensional systems with relevant nonlinearities, in such a way that, for the hydrodynamic noise, the infinite Onsager matrix (infinity being related to hydrodynamic superdiffusion) is projected to a subspace of the diffusive Hilbert space where it is finite.

**Acknowledgments.** I am grateful to Friedrich Hübner, Tomohiro Sasamoto and Mayank Sharma for discussions, and especially Takato Yoshimura for exchanging ideas while working on the simultaneous paper [39] where related results are obtained. I thank the Institute of Science Tokyo for hospitaly, and the workshop "Hydrodynamics of low-dimensional interacting systems: Advances, challenges, and future directions", YITP-T-25-03, at the Yukawa Institute for Theoretical Physics, where part of this work was done. This work was supported by EPSRC under grants EP/W010194/1, and EP/Z534304/1 (ERC advanced grant scheme).

## A  Hydrodynamic velocities and normal modes

We recall that derivatives with respect to conserved densities are related to covariances in the *macrocanonical* ensemble. For instance, by the chain rule, we find

$$\frac{\partial \mathsf{o}}{\partial \mathsf{q}_i} = \sum_j \mathsf{C}^{ij} \langle Q_j, o \rangle^{\mathsf{c}}_{\underline{\beta}} \tag{135}$$

where $\mathsf{C}^{ij}$ is the inverse of the covariance matrix $\mathsf{C}_{ij}$, that is $\sum_j \mathsf{C}_{ij} \mathsf{C}^{jk} = \delta_i{}^k$, and likewise

$$\frac{\partial^2 \mathsf{o}}{\partial \mathsf{q}_i \partial \mathsf{q}_k} = \sum_{jl} \mathsf{C}^{ij} \mathsf{C}^{kl} \langle Q_j, Q_l, o^- \rangle^{\mathsf{c}}_{\underline{\beta}}, \quad o^- := o - \sum_{ij} q_i \mathsf{C}^{ij} \langle Q_j, o \rangle^{\mathsf{c}}_{\underline{\beta}} \tag{136}$$

where

$$\langle Q_j, Q_l, o^- \rangle^{\mathsf{c}}_{\underline{\beta}} = \int \mathrm{dx dx}' \langle q_j(\mathrm{x}), q_l(\mathrm{x}'), o^-(0) \rangle^{\mathsf{c}}_{\underline{\beta}} \tag{137}$$

is an integrated three-point connected correlation function in stationary states called the *3-point coupling*.

The flux Jacobian $\mathsf{A}$ is the matrix obtained by differentiating the fluxes (average currents):

$$\frac{\partial \mathsf{j}_i}{\partial \mathsf{q}_l} = \mathsf{A}_i{}^l = \sum_k \mathsf{C}^{lk} \langle Q_k, j_i \rangle_{\underline{\beta}}^{\mathrm{c}}. \tag{138}$$

For the Hessian of the current, we denote

$$\mathsf{A}_i{}^{jk} = \frac{\partial^2 \mathsf{j}_i}{\partial \mathsf{q}_j \partial \mathsf{q}_k}. \tag{139}$$

The flux Jacobian satisfies

$$\mathsf{AC} = \mathsf{CA}^{\mathrm{T}} \tag{140}$$

as shown in various contexts and levels of generality in [73, 74, 23, 75, 41, 76, 33, 77]. This implies that each element of the flux Jacobian vector has real eigenvalues, because the matrix $\sqrt{\mathsf{C}}^{-1} \mathsf{A} \sqrt{\mathsf{C}}$, obtained by a similarity transformation, is real and symmetric. In particular

$$\mathrm{spec}\, \mathsf{A} = \{\mathsf{v}_I\}_I \subseteq \mathbb{R} \tag{141}$$

where we use the capital letter $I$ to indicate that it runs over the *normal mode basis*. The quantities $\mathsf{v}_I$ have the interpretation as the hydrodynamic velocities, or generalised sound velocities, for wave propagation. Using the orthogonal matrix $M$ diagonalising the symmetric matrix $\sqrt{\mathsf{C}}^{-1} \mathsf{A}(\hat{\boldsymbol{p}}) \sqrt{\mathsf{C}}$, the transformation to the normal modes can be chosen as $\mathsf{R} = M\sqrt{\mathsf{C}}^{-1}$:

$$q_I = \sum_j \mathsf{R}_I{}^j q_j, \quad \mathsf{R}\mathsf{A}\mathsf{R}^{-1} = \mathrm{diag}\,(\mathsf{v}_I)_I, \quad \mathsf{R}\mathsf{C}\mathsf{R}^{\mathrm{T}} = \mathbf{1}. \tag{142}$$

In particular, the Hessian in normal modes is

$$\mathsf{A}_i{}^{JK} = \langle Q_J, Q_K, j_i^- \rangle_{\underline{\beta}}^{\mathrm{c}} = \sum_{jk} (\mathsf{R}^{-1})_j{}^J (\mathsf{R}^{-1})_k{}^K \frac{\partial^2 \mathsf{j}_i}{\partial \mathsf{q}_j \partial \mathsf{q}_k}. \tag{143}$$

It is convenient to define the third cumulant, which is completely symmetric in its arguments,

$$\mathsf{C}_{ijk} = \frac{\partial \mathsf{q}_k}{\partial \beta^i \partial \beta^j} = \langle Q_i, Q_j, q_k \rangle_{\underline{\beta}}^{\mathrm{c}} \tag{144}$$

and in normal modes this is still a quantity that is generically non-diagonal and non-trivial,

$$\mathsf{C}_{IJK} = \sum_{ijk} ((\mathsf{R}^{-1})_i{}^I (\mathsf{R}^{-1})_j{}^J (\mathsf{R}^{-1})_k{}^K \mathsf{C}_{ijk}. \tag{145}$$

Clearly, by definition

$$\mathsf{A}_i{}^{JK} = \mathsf{A}_i{}^{KJ}. \tag{146}$$

It turns out, as shown in our separate work [43], that when the lower index $k$ is transformed to normal modes, there is a partial symmetry relation involving this index as well:

$$\mathsf{A}_I{}^{JK} - \mathsf{A}_J{}^{IK} = (\mathsf{v}_J - \mathsf{v}_I)\mathsf{C}_{IJK}, \tag{147}$$

which is a crucial relation for the consistency of the emergent fluctuating theory for 3-point correlation functions.

We refer to *velocity-diagonal 3-point couplings* and *diagonal 3-point couplings*, respectively, the quantities, here for the currents,

$$\mathsf{A}_k{}^{II} \quad \text{and} \quad \mathsf{A}_k{}^{IJ} \text{ for } I, J \text{ such that } \mathsf{v}_I = \mathsf{v}_J, \quad \text{respectively,} \tag{148}$$

and more generally for arbitrary observables

$$\langle Q_I, Q_I, o^- \rangle^{\mathrm{c}}_{\underline{\beta}} \quad \text{and} \quad \langle Q_I, Q_J, o^- \rangle^{\mathrm{c}}_{\underline{\beta}} \text{ for } I, J \text{ such that } \mathsf{v}_I = \mathsf{v}_J, \quad \text{respectively.} \tag{149}$$

The Euler scale hydrodynamic equation at large scales $\mathrm{x} = \ell x$, $\mathrm{t} = \ell t$, for local maximal entropy states parametrised by $\mathsf{q}_i(x, t)$, takes the form

$$\partial_t \mathsf{q}_i(x, t) + \sum_j \mathsf{A}_i{}^j(x, t) \partial_x \mathsf{q}_j(x, t) = 0. \tag{150}$$

In certain cases (but not all), it can be diagonalised by choosing appropriate, generically nonlinear functions $n_I = n_I(\underline{\mathsf{q}})$: if we can solve $\partial n_I / \partial \mathsf{q}_j = \mathsf{R}_I{}^j(\underline{\mathsf{q}})$, we obtain

$$\partial_t n_I + \mathsf{v}_I \partial_x n_I = 0. \tag{151}$$

The $n_I$'s are *Riemann invariants*, or "nonlinear normal modes". Note how the linear normal modes $q_I$ differ from $n_I$: the notation $q_I$ means the linear transformation (142), while $n_I$ is a nonlinear function of $\mathsf{q}_i$'s. At the linearised level the Euler equation can be diagonalised, and gives rise to the following simple Euler-scale correlation functions in the normal mode basis and for coarse-grained observables (see e.g. [33]),

$$\langle \overline{q_I}(x, t), \overline{q_J}(0, 0) \rangle^{\mathrm{c}}_{\underline{\beta}} \sim \ell^{-1} \delta(x - \mathsf{v}_I t) \delta_{I,J}. \tag{152}$$

or in the original basis

$$\langle \overline{q_i}(x, t), \overline{q_j}(0, 0) \rangle^{\mathrm{c}}_{\underline{\beta}} \sim \ell^{-1} (\delta(x - \mathsf{A}t)\mathsf{C})_{ij}. \tag{153}$$

# B  Initial conditions in the long-wavelength state

In this section all time coordinates are implicitly set to $t = 0$.

Consider the one-point function $\langle\!\langle q_i(x) \rangle\!\rangle_\ell = \langle \overline{q_i}(x) \rangle_\ell$. We first perform the calculation by omitting the fluid-cell mean for clarity. Expanding (12) for the one-point function $\langle q_i(\ell x) \rangle_\ell$, we get

$$\langle q_i(\ell x) \rangle_\ell = \langle q_i \rangle_{\underline{\beta}(x)} - \sum_j \frac{\partial_x \beta^j(x)}{\ell} \int \mathrm{dy}\, \mathrm{y} \langle q_i(0), q_j(\mathrm{y}) \rangle^{\mathrm{c}}_{\underline{\beta}(x)} + \mathcal{O}(\ell^{-2}). \tag{154}$$

By PT symmetry and translation invariance, we have

$$
\begin{aligned}
\int \mathrm{dy}\, \mathrm{y} \langle q_i(0), q_j(\mathrm{y}) \rangle^{\mathrm{c}}_{\underline{\beta}(x)} \quad &\overset{\text{PT symmetry}}{=} \quad \int \mathrm{dy}\, \mathrm{y} \langle q_i(0), q_j(-\mathrm{y}) \rangle^{\mathrm{c}}_{\underline{\beta}(x)} \\
&\overset{\text{change } \mathrm{y} \to -\mathrm{y}}{=} \quad -\int \mathrm{dy}\, \mathrm{y} \langle q_i(0), q_j(\mathrm{y}) \rangle^{\mathrm{c}}_{\underline{\beta}(x)} \\
&= \quad 0
\end{aligned} \tag{155}
$$

showing the first equation of (24). The fluid cell mean does not modify the first term in (154) by translation invariance; while in the second term the integral is modified to

$$\frac{1}{L}\int_{-L/2}^{L/2}\mathrm{dx}\int\mathrm{dy}\,(\mathrm{y}+\mathrm{x})\langle q_i(0),q_j(\mathrm{y})\rangle^{\mathrm{c}}_{\underline{\beta}(x)} \tag{156}$$

which, by PT symmetry is again zero. Note that it is essential that the fluid cell be *balanced*: the argument of the fluid-cell mean is the center of the segment over which we average (otherwise, there are nonzero contributions which are easily calculable).

For the two-point function in (24), by a scaling analysis we have in general

$$\langle\!\langle q_i(x)q_j(x')\rangle\!\rangle_\ell = \ell^{-1}\Big(\mathsf{C}_{ij}(x)+\ell^{-1}\delta^{(1)}\mathsf{C}_{ij}(x)\Big)\delta(x-x')+\ell^{-2}\delta^{(2)}\mathsf{C}_{ij}(x)\delta'(x-x')+\mathcal{O}(\ell^{-3})$$

for some regular functions $\delta^{(1,2)}\mathsf{C}_{ij}(x)$. We extract the $\delta(x-x')$ and $\delta'(x-x')$ by integrating against a slowly-varying smooth function of the microscopic coordinates. We do this on the correlation function in the original state $\langle\cdots\rangle_\ell$. Agian we first perform the calculation by omitting the fluid-cell mean for clarify. For smooth $g(y)$, we get

$$\begin{aligned}\int\mathrm{dy}\,g(\mathrm{y}/\ell)\,\langle q_i(\ell x)q_j(\ell x+\mathrm{y})\rangle^{\mathrm{c}}_\ell &= \int\mathrm{dy}\,g(\mathrm{y}/\ell)\,\langle q_i(0)q_j(\mathrm{y})\rangle^{\mathrm{c}}_{\underline{\beta}(x)}\\ &\quad -\sum_k\frac{\partial_x\beta^k(x)}{\ell}\int\mathrm{dydz}\,\mathrm{z}g(\mathrm{y}/\ell)\,\langle q_i(0)q_j(\mathrm{y})q_k(\mathrm{z})\rangle^{\mathrm{c}}_{\underline{\beta}(x)}+\mathcal{O}(\ell^{-2})\\ &= g(0)\mathsf{C}_{ij}(x)+\mathcal{O}(\ell^{-2})\end{aligned}\tag{157}$$

where for the first term in the parenthesis in the last step, we have used (155), and for the second term we have used a similar argument: PT-symmetry and the change $\mathrm{y},\mathrm{z}\longrightarrow-\mathrm{y},-\mathrm{z}$,

$$\mathsf{C}^{(1)}_{ijk}(x)=\int\mathrm{dydz}\,\mathrm{z}\,\langle q_i(0)q_j(\mathrm{y})q_k(\mathrm{z})\rangle^{\mathrm{c}}_{\underline{\beta}(x)}=-\int\mathrm{dydz}\,\mathrm{z}\,\langle q_i(0)q_j(\mathrm{y})q_k(\mathrm{z})\rangle^{\mathrm{c}}_{\underline{\beta}(x)}=0.\tag{158}$$

Therefore, $\delta^{(1)}\mathsf{C}_{ij}(x)=\delta^{.2)}\mathsf{C}_{ij}(x)=0$, and we recover the second equation of (24). The fluid cell mean must be applied to both observables. But again, because it is balanced, the PT-symmetry argument is not broken and the result remains true.

## C   No-shock systems: linear degeneracy

By definition, a hydrodynamic system in $d=1$ is linearly degenerate if each hydrodynamic velocity satisfies [26, 27, 28, 29, 30]

$$\sum_i(\mathsf{R}^{-1})_i{}^I\frac{\partial\mathsf{v}_I}{\partial\mathsf{q}_i}=0\quad\forall\,I.\tag{159}$$

If in addition the Riemann invariants $n_I$ exist, then this is

$$\frac{\partial\mathsf{v}_I}{\partial n_I}=0\quad\forall\,I.\tag{160}$$

In fact, each $\mathsf{v}_I$ may be said to be linearly degenerate or not [30]; here we use this terminology to mean that they all are.

For our purpose, the most crucial aspect of linearly degenerate systems is that, in the Euler-scale evolution from inhomogeneous state, no shock are produced [31, 32], see also [30, Sec 3.2, 3.3]. Integrable systems, described by Generalised Hydrodynamics (GHD), possess a continuum of modes $I \to \theta \in \mathbb{R}$, but a certain finite-mode reduction are linearly degenerate [28, 29], and in a formal sense even without reduction, they are linearly degenerate, see the lecture notes [33]. An independent proof of absence of shocks for a family of GHD equations is done in [34, 35].

Below, we will not use (159). Instead, we use, in a subtle but crucial way, the absence of shock production. Thus, our results hold for no-shock systems, of which linearly degenerate systems are examples (perhaps the only ones).

## C.1  Proof that the velocity-diagonal 3-point couplings vanish

Here we reproduce the main lines of the calculation in [25, Sup Mat], Section "Degenerate three-point coupling from linear degeneracy". As far as we are aware, this calculation is the first proof of the vanishing of the diagonal 3-point coupling in no-shock systems. We provide here more details, rendering slightly more explicit tacit assumptions that were made, and pointing out steps that would require a more detailed analysis of correlation functions decay.

We show that, in no-shock systems,

$$\mathsf{A}_k^{IJ} = 0 \quad \text{whenever } \mathsf{v}_I = \mathsf{v}_J \tag{161}$$

and more generally

$$\langle Q_I, Q_J, o^- \rangle^{\mathrm{c}}_{\underline{\beta}} = 0 \quad \text{whenever } \mathsf{v}_I = \mathsf{v}_J \tag{162}$$

for every local observable $o$.

This does not appear to follow from the Euler-scale calculus of linear degeneracy – that is, it does not follow by direct calcualtion, by taking derivatives and going to the normal mode basis, from the basic definition (159). Instead, it requires us to consider the statistical mechanical model underlying the hydrodynamic equation. Thus, it is an extra constraint on the hydrodynamic equation for it to emerge from a many-body system, in the case where it is linearly degenerate.

*Proof.* Consider a stationary state

$$\langle \cdots \rangle = \langle \cdots \rangle_{\underline{\beta}} \tag{163}$$

and the quantity

$$U(o_1, o_2) = \lim_{T \to \infty} \frac{1}{T} \int_0^T \mathrm{dt} \int \mathrm{dx} \, \langle o_1(\mathrm{x}, \mathrm{t}), o_2^-(0, 0) \rangle^{\mathrm{c}}. \tag{164}$$

We recall the notation

$$o^-(\mathrm{x}, \mathrm{t}) = o(\mathrm{x}, \mathrm{t}) - \sum_{ij} q_i(\mathrm{x}, \mathrm{t}) \mathsf{C}^{ij} \langle Q_j, o \rangle. \tag{165}$$

By the general projection result [51, Thm 6.1], we have

$$U(o_1, o_2) = 0 \tag{166}$$

for every local observable $o_1, o_2$.

Consider

$$\langle\!\langle q_I(x,0), q_J(y,0), o^-(z,t)\rangle\!\rangle^{\mathrm{c}}. \tag{167}$$

By the nonlinear projection result for three-point function [8] (see also [43]), we have

$$\langle\!\langle q_I(x,0), q_J(y,0), o^-(z,t)\rangle\!\rangle^{\mathrm{c}} = \sum_{ij} \frac{\partial^2 \mathsf{o}}{\partial \mathsf{q}_i \partial \mathsf{q}_j} \langle\!\langle q_I(x,0)q_i(z,t)\rangle\!\rangle^{\mathrm{c}} \langle\!\langle q_J(x,0)q_j(z,t)\rangle\!\rangle^{\mathrm{c}}. \tag{168}$$

As explained in [43], with the absence of shock production, projection results hold for all values of space-time. Hence, in no-shock systems, the above holds for all space-time, and we can use the explcit solution for two-point functions. We pass to normal modes using the first equation in (162), obtaining the result first established in [24, App E]:

$$\langle\!\langle q_I(x,0), q_J(y,0), o^-(z,t)\rangle\!\rangle^{\mathrm{c}} = \ell^{-2}\langle Q_I, Q_J, o^-\rangle^{\mathrm{c}}\delta(z - x - \mathsf{v}_I t)\delta(z - y - \mathsf{v}_J t). \tag{169}$$

With $q_I^0 = q_I - \langle\!\langle q_I \rangle\!\rangle \mathbf{1}$, at $z = 0$, and after replacing $t \to -t$, we have a result for the following second cumulant:

$$\langle\!\langle q_I^0(x,t)q_J^0(y,t), o^-(0,0)\rangle\!\rangle^{\mathrm{c}} = \ell^{-2}\langle Q_I, Q_J, o^-\rangle^{\mathrm{c}}\delta(x - \mathsf{v}_I t)\delta(y - \mathsf{v}_J t). \tag{170}$$

Re-interpreting this in terms of microscopic coordinates in the full stationary state, this is

$$\langle q_I^0(\mathsf{x},\mathsf{t})q_J^0(\mathsf{y},\mathsf{t}), o^-(0,0)\rangle^{\mathrm{c}} = \langle Q_I, Q_J, o^-\rangle^{\mathrm{c}}f(\mathsf{x},\mathsf{y},\mathsf{t}). \tag{171}$$

Here $f$ is finite and supported on $(\mathsf{x},\mathsf{y}) \in [\mathsf{v}_I\mathsf{t} - L, \mathsf{v}_I\mathsf{t} + L] \times [\mathsf{v}_J\mathsf{t} - L, \mathsf{v}_J\mathsf{t} + L]$ for some $L$, plus corrections, outside this region, that vanish "weakly on large scales", that is, under integration with slowly-varying smooth functions $g(\mathsf{x}/\ell, \mathsf{y}/\ell)$ after taking $\mathsf{t} = \ell t$ with $\ell_{\mathrm{micro}} \ll L \ll \ell \to \infty$. We denote the total y integral, which is finite, as

$$F(\mathsf{x},\mathsf{t}) := \int \mathrm{d}\mathsf{y}\, f(\mathsf{x},\mathsf{y},\mathsf{t}) \tag{172}$$

and note that it integrates to $\int \mathrm{d}\mathsf{x}\, F(\mathsf{x},\mathsf{t}) = 1$.

Now take a function $w(z)$ that is even, bounded, smooth and supported on $[-1,1]$, with $w(0) = 1$, and define

$$o'(\mathsf{x},\mathsf{t}) = \int \mathrm{d}\mathsf{y}\, w((\mathsf{y} - \mathsf{x})/\ell)q_I^0(\mathsf{x},\mathsf{t})q_J^0(\mathsf{y},\mathsf{t}). \tag{173}$$

For all finite $\ell > 0$, this is a local observable, as it has a finite support. Using (171) and the information about the support of $f$, we find, for every smooth functions $g$,

$$\int \mathrm{d}\mathsf{x}\, g(\mathsf{x}/\ell)\langle o'(\mathsf{x},\ell t), o^-(0,0)\rangle^{\mathrm{c}} = \langle Q_I, Q_J, o^-\rangle^{\mathrm{c}}w((\mathsf{v}_I - \mathsf{v}_J)t) \int \mathrm{d}\mathsf{x}\, g(\mathsf{x}/\ell)F(\mathsf{x}, \ell t) + o(\ell_{\mathrm{micro}}/L, L/\ell). \tag{174}$$

Taking $g = 1$ and averaging over $t \in [0, T]$, we therefore obtain

$$\frac{1}{\ell T} \int_0^{\ell T} \mathrm{d}t \int \mathrm{d}\mathsf{x}\, \langle o'(\mathsf{x},\mathsf{t}), o^-(0,0)\rangle^{\mathrm{c}} = \langle Q_I, Q_J, o^-\rangle^{\mathrm{c}}R(T) + o(\ell_{\mathrm{micro}}/L, L/\ell) \tag{175}$$

where

$$R(T) = \begin{cases} 1 & (\mathsf{v}_I = \mathsf{v}_J) \\ \dfrac{1}{|\mathsf{v}_I - \mathsf{v}_J|T} \displaystyle\int_0^{\min(1,|\mathsf{v}_I - \mathsf{v}_J|T)} \mathrm{d}z\, w(z) & \text{(otherwise)}. \end{cases} \tag{176}$$

Taking the limit $\lim_{T\to\infty}$, we get

$$U(o',o) = \begin{cases} \langle Q_I, Q_J, o^- \rangle^{\mathrm{c}} & (\mathsf{v}_I = \mathsf{v}_J) \\ 0 & \text{(otherwise)} \end{cases} + o(\ell_{\mathrm{micro}}/L, L/\ell). \tag{177}$$

From the general result (166), and as we can take $L, \ell$ as large as desired with $L \ll \ell$, we have shown the vanishing of the 3-point coupling. ∎

## C.2 Proof that the short-range two-point correlation in a large-wavelength non-equilibrium state is given by the equilibrium covariance matrix

This proof is based on the calculation found in [25, Sup Mat], Section "Dynamics of two-point correlations". This calculation as it is, is not enough to derive the result we want here, so we complete it. Note that the result was claimed in [8] using BMFT for integrable systems, however the poof there was not complete, as the commutant of the flux Jacobian was not considered.

For simplicity we assume that hydrodynamic velocities are non-degenerate $\mathsf{v}_I \neq \mathsf{v}_{I'} \,\forall\, I \neq I'$ (for generic states). That is, the hydrodynamic equation is strictly hyperbolic.

Consider the equal-time two-point Euler amplitude $S_{ij}(x, x'; t) := \lim_{\ell\to\infty} \ell \langle\!\langle q_i(x,t), q_j(x',t) \rangle\!\rangle_\ell^{\mathrm{c}}$ of conserved densities in a long-wavelength inhomogeneous state, at the Euler scale. The initial condition (24) is

$$S_{ij}(x, x'; 0) = \mathsf{C}_{ij}(x, 0)\delta(x - x') \tag{178}$$

where, as above, we use the notation $\mathsf{C}_{ij}(x, t) = \mathsf{C}_{ij}(\underline{\mathsf{q}}(x, t))$ for the equilibirum covariance matrix in the state determined by the $\mathsf{q}_i(x,t)$'s, Eq. (9).

We will show that for all times,

$$S_{ij}(x, x'; t) = \mathsf{C}_{ij}(x, t)\delta(x - x') + \text{regular} \tag{179}$$

*Proof.* $S_{ij}(x, x'; t)$ satisfies the Euler-scale propagation equation

$$\partial_t S_{ij}(x, x', t) + \sum_k \partial_x \Big(\mathsf{A}_i{}^k(x, t) S_{kj}(x, x', t)\Big) + \sum_k \partial_{x'} \Big(S_{ik}(x, x', t)\mathsf{A}_j{}^k(x', t)\Big) = 0 \tag{180}$$

where $\mathsf{A}(x, t) = \mathsf{A}(\underline{\mathsf{q}}(x, t))$ is the flux Jacobian evaluated at the stationary state charactersied by $\mathsf{q}_i(x, t)$'s. This follows from the BMFT [7, 8]. Equivalently, it follows from Euler-scale projections, that is the nonlinear Boltzmann Gibbs principle Eq. (21) without bare diffusion and noise, along with the fact that no shocks are produced, so that the projection holds everywhere in space-time [43].

The general solution structure is

$$S_{ij}(x, x', t) = \tilde{C}_{ij}(x, t)\delta(x - x') + E_{ij}(x, x', t) \tag{181}$$

where $\tilde{C}_{ij}(x,t)$ and $E_{ij}(x,x',t)$ are ordinary functions. In particular $\tilde{C}_{ij}(x,t)$ is not assumed to be a function of the local state determined by $\mathsf{q}_i(x,t)$'s. By integrating $\langle\!\langle q_i(x,t), q_j(x',t)\rangle\!\rangle_\ell^{\mathrm{c}}$ on $x, x' \in [-\epsilon, \epsilon]$ and dividing by $\epsilon$, we obtain a non-negative quantity, the local covariance of conserved quantities. As conserved quantities must be non-degenerate (Sec. 2), the result must in fact be strictly positive, hence $\tilde{C} > 0$.

In (180), the form (181) leads to a term proportional to $\delta'(x - x')$, which turns out to have the form [25, Sup Mat] (see Section "Dynamics of two-point correlations")

$$\left(\mathsf{A}(x,t)\tilde{C}(x,t) - \tilde{C}(x,t)\mathsf{A}(x,t)^{\mathrm{T}}\right)_{ij} \delta'(x - x'). \tag{182}$$

As this must vanish, we must have $\mathsf{A}(x,t)\tilde{C}(x,t) = \tilde{C}(x,t)\mathsf{A}(x,t)^{\mathrm{T}}$. We may always write

$$\tilde{C}(x,t) = \mathsf{R}(x,t)^{-1} m(x,t) \mathsf{R}(x,t)^{-\mathrm{T}} \tag{183}$$

for some positive matrix $m(x,t) > 0$, where $\mathsf{R}(x,t) = \mathsf{R}(\mathsf{q}(x,t))$ is the local normal-mode transformation matrix, Eq. (142). Thus we obtain the condition

$$[\mathrm{diag}(\mathsf{v}_I(x,t))_I, m(x,t)] = 0. \tag{184}$$

Under the assumption that the hydrodynamic velocities are non-degenerate $\mathsf{v}_I(x,t) \neq \mathsf{v}_{I'}(x,t) \, \forall I \neq I'$, we must have $m(x,t) = \mathrm{diag}(m(x,t)_I)_I$ with $m(x,t)_I > 0$. Because of (178), we also have

$$m(x,0)_I = 1 \quad \forall I. \tag{185}$$

For lightness of notation, from now on in this proof we do not write the explicit $x, t$ dependence of matrices. Let us construct the matrix $M(x,t)$ as

$$M(x,t) = \mathsf{R}(x,t)^{-1} m(x,t) \, \mathsf{R}(x,t). \tag{186}$$

hence we have

$$\tilde{C} = \mathsf{R}^{-1} m \, \mathsf{R}^{-\mathrm{T}} = M\mathsf{C} = \mathsf{C}M^{\mathrm{T}}. \tag{187}$$

The term proportional to $\delta(x - x')$ coming from $\tilde{C}_{ij}\delta(x - x')$ is

$$\left(\partial_t \tilde{C} + \partial_x(\mathsf{A}\tilde{C})\right)_{ij} \delta(x - x'). \tag{188}$$

It is shown in [25, Sup Mat] that, with the standard definition $\mathsf{B} := \mathsf{A}\mathsf{C}$,

$$\left(\partial_t \mathsf{C} + \partial_x \mathsf{B}\right)_{ij} = \langle Q_i, Q_j, j_k^- \rangle^{\mathrm{c}} \mathsf{C}^{kl} \partial_x \mathsf{q}_l \tag{189}$$

thus the term (188) is

$$\left(\mathsf{C}\partial_t M^{\mathrm{T}} + \mathsf{B}\partial_x M^{\mathrm{T}} + \sum_{kl} \langle Q_\cdot, Q_\cdot, j_k^- \rangle^{\mathrm{c}} \mathsf{C}^{kl} \partial_x \mathsf{q}_l \right)_{ij} \delta(x - x'). \tag{190}$$

So we must have

$$\partial_t E_{ij} + \sum_k \partial_x \left(\mathsf{A}_i{}^k E_{kj}\right) + \sum_k \partial_{x'} \left(E_{ik} \mathsf{A}_j{}^k\right)$$
$$= -\left(\mathsf{C}\partial_t M^{\mathrm{T}} + \mathsf{B}\partial_x M^{\mathrm{T}} + \sum_{kl} \langle Q_\cdot, Q_\cdot, j_k^- \rangle^{\mathrm{c}} \mathsf{C}^{kl} \partial_x \mathsf{q}_l \right)_{ij} \delta(x - x'). \tag{191}$$

On the left-hand side, only derivatives in $x, x'$ may produce a delta-function $\delta(x - x')$, and this, because of discontinuities. These can only come from $E_{ij}(x, x', t)$, and using the basis of derivatives $\partial_\pm = \partial_x \pm \partial_{x'}$, can only come from $\partial_- E_{ij}$. Passing to normal modes, and taking into consideration that neither $\mathsf{A}$ nor the transformation matrix $\mathsf{R}$ can produce delta-functions $\delta(x - x')$, we get

$$\mathsf{v}_I \partial_x E_{IJ} + \mathsf{v}_J \partial_{x'} E_{IJ} \sim -\Big( (\mathsf{R} \partial_t M \mathsf{R}^{-1} + \mathsf{v}_I \mathsf{R} \partial_x M \mathsf{R}^{-1})_J^I + \sum_{kl} \langle Q_I, Q_J, j_k^- \rangle^{\mathrm{c}} \mathsf{C}^{kl} \partial_x \mathsf{q}_l \Big) \delta(x - x') \quad (192)$$

For $I = J$, and using the fact that $\langle Q_I, Q_J, j_K^- \rangle^{\mathrm{c}} = 0$ whenever $\mathsf{v}_I = \mathsf{v}_J$ (Sec. (C.1)), we obtain

$$\mathsf{v}_I (\partial_x + \partial_{x'}) E_{II} \sim -(\mathsf{R} \partial_t M \mathsf{R}^{-1} + \mathsf{v}_I \mathsf{R} \partial_x M \mathsf{R}^{-1})_I^I \, \delta(x - x'). \quad (193)$$

As the left-hand side cannot produce $\delta(x - x')$, the coefficient of the delta function on the right-hand side must vanish. This is

$$\Big( \partial_t m + \mathsf{v}_I \partial_x m + [m, (\partial_t \mathsf{R} + \mathsf{v}_I \partial_x \mathsf{R}) \mathsf{R}^{-1}] \Big)_I^I = 0. \quad (194)$$

Because $m$ is diagonal, the second term, the commutator, evaluated on the diagonal $(\cdot)_I^I$, vanishes. Thus we have

$$\partial_t m_I + \mathsf{v}_I \partial_x m_I = 0 \ \forall I. \quad (195)$$

Eq. (195) says that $m_I$ is transported along the characteristics of the fluid. Because there are no shocks, all characteristics come from the initial time. Because of the initial condition (185), we then have

$$m_I(x, t) = 1. \quad (196)$$

∎

# D    Drude weight from nonlinear fluctuating Boltzmann-Gibbs principle

Recall that as a general theorem [51], the Drude weight is given by the projection onto the conserved quantities:

$$\mathsf{D}_{o_1, o_2} = \frac{\partial \mathsf{o}_1}{\partial \mathsf{q}_{i_1}} \frac{\partial \mathsf{o}_2}{\partial \mathsf{q}_{i_2}} \mathsf{C}_{i_1, i_2}. \quad (197)$$

It is instructive to verify that our formalism is in agreement with the general result (197), by evaluating the Drude weight using the nonlinear fluctuating Boltzmann-Gibbs principle (21).

Consider the definition (46) written in terms of macrosocpic coordinates for fluid-cell means in the fluctuation theory,

$$\mathsf{D}_{o_1, o_2} = \lim_{t \to \pm\infty} \ell \int \mathrm{d}x \, \langle\!\langle o_1(x, t), o_2(0, 0) \rangle\!\rangle^{\mathrm{c}}. \quad (198)$$

Consider $\langle\!\langle o_1(x_1, t_1), o_2(x_2, t_2) \rangle\!\rangle^{\mathrm{c}}$, which depends on $x_1 - x_2$ and $t_1 - t_2$ by space-time translation invariance. We may integrate over $x_1$, or integrate over $x_2$, and we look at the limit $t_1 - t_2 \to \pm\infty$.

Clearly, the noise-boise term (71) gives zero under the limit $t_1 - t_2 \to \pm\infty$, and by the results of Sec. 4.2 all other correlations vanish except for that from the microcanonical terms in (21), that is (65).

For the first line on the right-hand side of (65), we obtain (197). Hence, we need to verify that the other lines give zero.

For the last line on the right-hand side of (65), we integrate over $x_1$ and notice that the result is independent of $t_1, t_2$:

$$\int dx_1 \, \langle\!\langle q_{i_1}(x_1, t_1), q_{i_2}(x_2 \pm 0^+, t_2), q_{i_2'}(x_2 \mp 0^+, t_2) \rangle\!\rangle^{\mathrm{c}} \tag{199}$$
$$= \quad \langle\!\langle Q_1, q_{i_2}(x_2 \pm 0^+, t_2), q_{i_2'}(x_2 \mp 0^+, t_2) \rangle\!\rangle^{\mathrm{c}} = -\frac{\partial}{\partial\beta^1} \langle\!\langle q_{i_2}(\pm 0^+, 0), q_{i_2'}(\mp 0^+, 0) \rangle\!\rangle^{\mathrm{c}} = 0$$

where we use the fact that the state has short-range correlations. Similarly, we obtain 0 for the third line on the right-hand side of (65) by integrating over $x_2$.

For the second line on the right-hand side of (65), we set $x_1 = x, t_1 = t$ and $x_2 = t_2 = 0$, and, as in Sec. 4.2, we pass to normal modes using

$$\frac{\partial^2 \mathsf{o}_1}{\partial \mathsf{q}_{i_1} \partial \mathsf{q}_{i_1'}} = \langle o_1^-, Q_{j_1}, Q_{j_1'} \rangle^{\mathrm{c}} \mathsf{C}^{j_1, i_1} \mathsf{C}^{j_1', i_1'} \tag{200}$$

and the normal-mode transformation (142), to obtain

$$\langle o_1^-, Q_{I_1}, Q_{I_1'} \rangle^{\mathrm{c}} \langle o_2^-, Q_{I_2}, Q_{I_2'} \rangle^{\mathrm{c}} \langle\!\langle q_{I_1}(x, t), q_{I_2}(0, 0) \rangle\!\rangle^{\mathrm{c}} \langle\!\langle q_{I_1'}(x, t), q_{I_2'}(0, 0) \rangle\!\rangle^{\mathrm{c}}. \tag{201}$$

This can be evaluated exactly to its leading $\mathcal{O}(\ell^{-2})$ order by using the Euler-scale formula for the two-point functions

$$\ell^{-2} \frac{1}{2} \langle o_1^-, Q_I, Q_{I'} \rangle^{\mathrm{c}} \langle o_2^-, Q_I, Q_{I'} \rangle^{\mathrm{c}} \delta(x - \mathsf{v}_I t) \delta(x - \mathsf{v}_{I'} t). \tag{202}$$

where we used (64) for the projected observable $o^-$. Clearly, the integral over $x$ is 0 if for every values of the indices $I, I'$ we have either $\mathsf{v}_I \neq \mathsf{v}_{I'}$, or $\langle o_1^-, Q_I, Q_{I'} \rangle^{\mathrm{c}} = 0$, or $\langle o_2^-, Q_I, Q_{I'} \rangle^{\mathrm{c}}$. As we assume linear degeneracy, (42) holds, hence the result vanishes.

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
