# Peer review of "Hydrodynamic noise in one dimension: projected Kubo formula and how it vanishes in integrable models"

_SciPost Physics_

## Round 2 · Referee Report · Anonymous (Referee 2) · 2025-10-31

Strengths

1) Determines the exact corrections to the deterministic hydrodynamic (large scale) theory for a large class of 1D interacting system.

2) Physically discusses the origin of noise and characterises its covariance in terms of a new "projected Kubo formula".

3) Identifies the diffusive corrections and derives an extended Einstein relation that links the noise covariance to the bare diffusion coefficients.

4) Proves a recent conjecture that noise vanishes for integrable systems.

Weaknesses

The notations can be difficult to follow. For instance, understanding all the equalities in an equation like (53) currently requires to go look for the different definitions that are scattered through the article.

Report

This is a very interesting work, that brings new mathematical and physical understanding to the emergence of hydrodynamic theory (including its corrections) at large scales. It contains a lot a results and thus takes a lot of time to read, but it is definitely worth reading. The article contains very interesting physical discussions, in particular in Section 3, which greatly help understanding the physics behind the equations. It definitely meets the criteria for publication in SciPost Physics. Nevertheless, I think the readability can be increased easily (in particular in regards of the numerous notations), see below.

Minor comments/suggestions:

1) The equivalence between microcanonical and canonical ensembles is used several times. But for finite systems of length L (size of the fluid cell), one expects corrections. Are these corrections subleading and thus irrelevant? Could the author add a comment on this point?

2) On page 7, around Eq. (10): why is the fact that the map $\underline{\beta} \to \langle \underline{q} \rangle_{\beta}$ is invertible related to the positivity covariance matrix?

3) In Eq. (14), the author could clarify the meaning of $\mathcal{O}(\ell^{-\infty})$ which is used here for the first time and can be a bit surprising, in particular since there is also a $\mathcal{O}(\ell^{-2})$ in the same equation.

Requested changes

1) Include a summary of the notations and definitions used in the article (for the observables, averagings, ...). That would greatly help the reader.

2) typos: - p. 10, after (27) "noise, They" $\to$ "noise. They" - p. 12, "conserve density" $\to$ "conserved density" - p. 12, "Physcailly" $\to$ "Physically" - p. 13, "not bee too large" $\to$ "not be too large" - p. 25, "intregable" $\to$ "integrable" - p. 26, "scatteing" $\to$ "scattering" - p. 29, "ligthness" $\to$ "lightness" - p. 33, "disccused" $\to" "discussed" - p. 33, "via because of"

Recommendation

Publish (easily meets expectations and criteria for this Journal; among top 50%)

---

## Round 2 · Referee Report · Anonymous (Referee 1) · 2025-10-31

Report

This paper is concerned with the noise in the fluctuating hydrodynamics of integrable systems. It provides several proofs that the bare noise is absent in these systems (although there are indirect mechanisms, uncovered in previous work, that effectively produce noise through nonlinear coupling to other modes). This confirms previous conjectures made by the author and others.

The paper is well written, with an emphasis on precision at the cost of readability. I only have a few clarifying questions:

  • The concept of "linearly degenerate systems" plays an important role throughout. They seem to be defined as systems where shocks are not produced at the Euler scale. This implies no cubic nonlinearity and hence no emergence of KPZ. Does this also imply no quartic nonlinearity? These can arise in systems with particle-hole symmetry (see, e.g., [36]), which are still marginal terms in the fluctuating hydrodynamics, and thus lead to logarithmic corrections to diffusion.

  • Above Eq. (6): I suspect the consequence of PT symmetry on diffusive fluctuating hydrodynamics long predates [41] - for example in textbooks by Forster, or Chaikin & Lubensky, or others. It is also pervasive in part of the recent litterature on fluctuating hydrodynamics.

  • "We assume that β i ’s are away from phase transitions, so that correlations are short-range in these states" - is it then also assumed that we are not in an ordered phase?

Recommendation

Publish (easily meets expectations and criteria for this Journal; among top 50%)

---

## Editorial Decision

awaiting_resubmission